# Phosphorylation, disorder, and phase separation govern the behavior of Frequency in the fungal circadian clock

Daniyal Tariq[1†], Nicole Maurici[2†], Bradley M Bartholomai[3],
Siddarth Chandrasekaran[1], Jay C Dunlap[3], Alaji Bah[2]*, Brian R Crane[1]*

[1]Department of Chemistry & Chemical Biology, Cornell University, Ithaca, United States; [2]Department of Biochemistry and Molecular Biology, SUNY Upstate Medical University, Syracuse, United States; [3]Department of Molecular and Systems Biology, Geisel School of Medicine at Dartmouth, Hanover, United States

*For correspondence:
baha@upstate.edu (AB);
bc69@cornell.edu (BRC)

[†]These authors contributed equally to this work

Competing interest: The authors declare that no competing interests exist.

**Abstract** Circadian clocks are composed of transcription-translation negative feedback loops that pace rhythms of gene expression to the diurnal cycle. In the filamentous fungus *Neurospora crassa*, the proteins **F**requency (**F**RQ), the **F**RQ-interacting RNA helicase (FRH), and **C**asein-Kinase I (CK1) form the **FFC** complex that represses expression of genes activated by the white-collar complex (WCC). FRQ orchestrates key molecular interactions of the clock despite containing little predicted tertiary structure. Spin labeling and pulse-dipolar electron spin resonance spectroscopy provide domain-specific structural insights into the 989-residue intrinsically disordered FRQ and the FFC. FRQ contains a compact core that associates and organizes FRH and CK1 to coordinate their roles in WCC repression. FRQ phosphorylation increases conformational flexibility and alters oligomeric state, but the changes in structure and dynamics are non-uniform. Full-length FRQ undergoes liquid–liquid phase separation (LLPS) to sequester FRH and CK1 and influence CK1 enzymatic activity. Although FRQ phosphorylation favors LLPS, LLPS feeds back to reduce FRQ phosphorylation by CK1 at higher temperatures. Live imaging of *Neurospora* hyphae reveals FRQ foci characteristic of condensates near the nuclear periphery. Analogous clock repressor proteins in higher organisms share little position-specific sequence identity with FRQ; yet, they contain amino acid compositions that promote LLPS. Hence, condensate formation may be a conserved feature of eukaryotic clocks.

## eLife assessment

This article is a **fundamental** contribution to the understanding of the role of intrinsically disordered proteins in circadian clocks and the potential involvement of phase separation mechanisms. The authors **convincingly** report on the structural and biochemical aspects and the molecular interactions of the intrinsically disordered protein FRQ. The article will be of interest to scientists focusing on circadian clock regulation, liquid–liquid phase separation, and phosphorylation.

## Introduction

Eukaryotic circadian rhythms of biological phenomena arise from periodic gene expression patterns that oscillate independent of external cues but can be entrained to light and temperature (*Hurley et al., 2016*). At the molecular level, circadian clocks function as transcription-translation negative feedback loops (TTFLs) that consist of core positive and negative elements (*Figure 1A*; *Crane and Young, 2014*). In the long-standing model clock of *Neurospora crassa*, the transcriptional repressor

**eLife digest** Natural oscillations known as circadian rhythms influence many processes in humans and other animals including sleep, eating, brain activity and body temperature. These rhythms allow us to anticipate and prepare for regular changes in our environment including day-night cycles and the temperature of our surroundings.

Circadian clocks in animals, fungi and other 'eukaryotic' organisms rely on networks of components that repress their own production to generate oscillations in their levels in cells over the course of a 24-hour period. The components in animal and fungus circadian clocks are different but there are strong similarities in their properties and how the networks operate. As a result, a type of fungus known as *Neurospora crassa* is often used as a model to study how circadian rhythms work in animals.

A central component in the *N. crassa* circadian clock is a protein known as Frequency (FRQ). It is a large protein that, unlike most proteins, lacks a well-defined, three-dimensional structure. Despite this, it is able to bind to and regulate other proteins to repress its own production. One of its protein partners known as CK1 attaches small tags known as phosphate groups to FRQ to set the length of the circadian rhythm. However, it remains unclear how FRQ interacts with its protein partners or what effect the phosphate groups have on its activity.

To address this question, Tariq, Maurici et al. used biochemical approaches to study the structure of FRQ. The experiments revealed that it contains a compact core that is able to bind to CK1 and other protein partners. The way FRQ regulates its protein partners is unusual: it undergoes a chemical process known as liquid-liquid phase separation to sequester other circadian clock proteins and modulate their enzymatic activities. In this process, a solution containing molecules of FRQ separates into two distinct components (known as phases), one of which contains FRQ and its partners in a concentrated liquid-like mixture. Evidence for such mixtures has also been found in living fungal cells.

Further experiments suggest that liquid-liquid phase separation of FRQ may allow the clock to compensate for changes in temperature to maintain a regular rhythm. The circadian clocks of animals and other organisms all have proteins that perform similar roles as FRQ and maintain sequence properties that promote liquid-liquid phase separation. Therefore, it is possible that liquid-liquid phase separation may be a common feature of circadian rhythms in nature.

complex is known as the FFC, named for its three principal components: Frequency (FRQ, *Figure 1B*), the FRQ-interacting RNA helicase (FRH), and Casein Kinase 1 (CK1) (*Cha et al., 2015*; *Hurley et al., 2016*). FRQ is functionally analogous to Period (PER) in insects and mammals, whereas CK1 is an ortholog of the mammalian CK1 and the *Drosophila* kinase Doubletime (DBT) (*Dunlap and Loros, 2017*). FRH is found only in the core repressor of the fungal clock. Although FRH is homologous to the yeast helicase MTR4, FRH helicase activity does not appear essential to clock function (*Conrad et al., 2016*; *Hurley et al., 2013*). The FFC represses the positive arm of the cycle, which involves two zinc-finger transcription factors, white-collar 1 (WC-1) and white-collar 2 (WC-2) assembled into the white-collar complex (WCC).

WCC binding to the clock-box (c-box) activates *frq* expression (*Figure 1A*; *Froehlich et al., 2002*). FRQ production is also highly regulated at the post-transcriptional level through the use of rare codons to retard translation, the expression of an antisense transcript and alternative splicing mechanisms (*Cha et al., 2015*; *Dunlap and Loros, 2017*; *Zhou et al., 2013*). The latter leads to the production of either a long (L-FRQ) or short (S-FRQ) isoform, and the ratio of these isoforms fine-tunes period length in response to ambient temperatures (*Diernfellner et al., 2007*; *Liu et al., 1997*; *Zhou et al., 2013*). Translated FRQ binds to FRH via its FFD domain and to CK1 via its two FCD domains (1 and 2) to form the repressive FFC (*Cha et al., 2015*; *Dunlap and Loros, 2017*; *Jankowski et al., 2022*; *Figure 1C*), although short linear motifs within FRQ may also broadly influence the FRQ-interactome (*Pelham et al., 2021*). The FFC enters the nucleus where it directly inhibits the WCC and thereby downregulates *frq* expression (*Hurley et al., 2013*). FRQ-mediated WCC phosphorylation by CK1 (and associated kinases) reduces interactions between the WCC and the c-box to close the feedback loop and restart the cycle (*Figure 1A*; *Wang et al., 2019*). FRH facilitates interactions between the FFC and the WCC (*Conrad et al., 2016*), and thus, repressor function likely depends upon the FRQ-mediated association of CK1 and FRH. Whereas interaction regions on the FRQ protein sequence

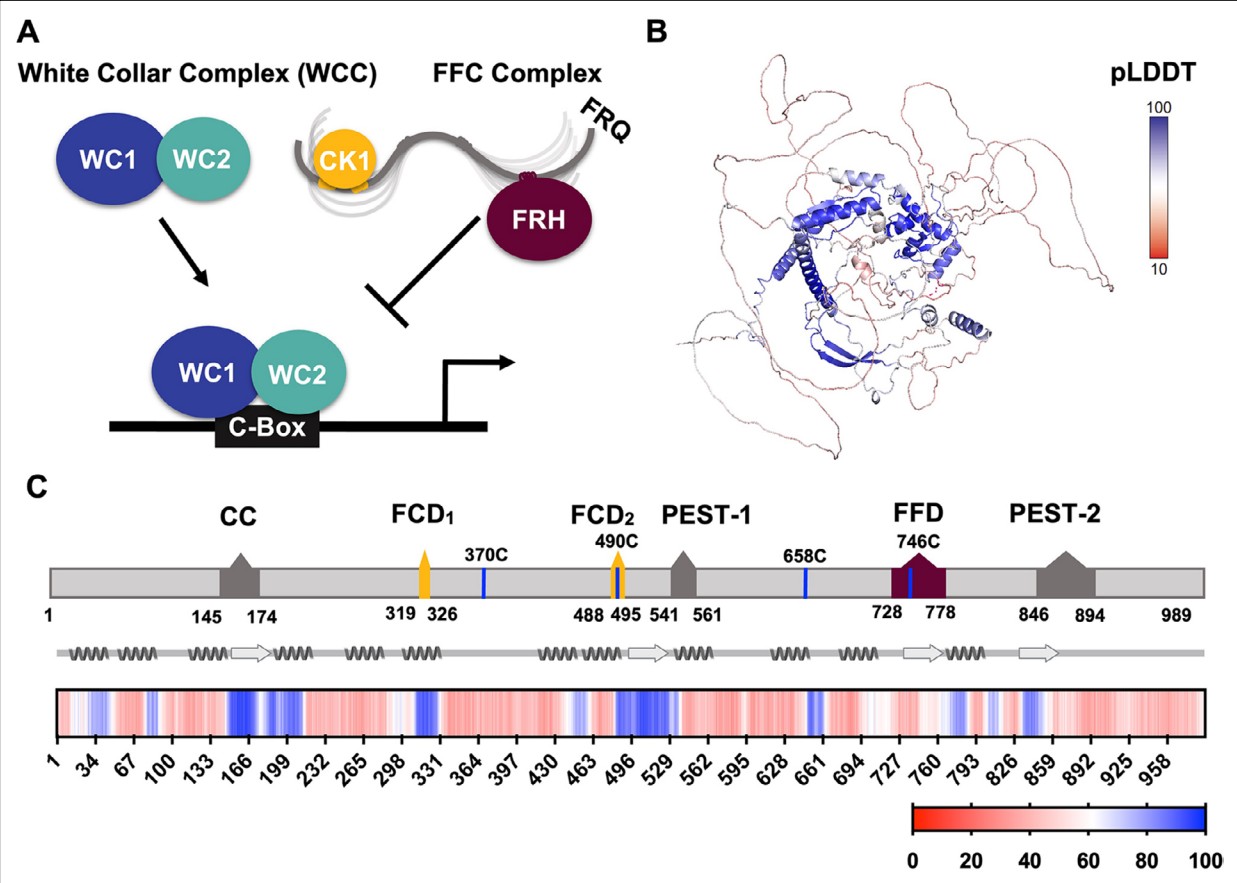

**Figure 1.** FRQ is a large, dynamic component of the clock oscillator. (**A**) LLPS propensity predictions using Pi-Pi and IUPred2A disorder predictions are shown for homologs of FRQ (n=47) (purple), dPER (n=36) (teal), and hPER1 (n=57) (orange) compared to values for proteins of similar length from their respective proteomes (Prot): Nc, *Neurospora crassa* (n=46); Dc, *Drosophila melanogaster* (n=121); Hc, Homo sapiens (n=626). This complex inhibits the positive-acting transcription factor known as the white-collar complex (WCC), composed of white collar-1 (WC-1) and white collar-2 (WC-2). The WCC activates *frq* expression by binding to the c-box region in the *frq* promoter. Stoichiometries of the components are not represented in the schematic. (**B**) AlphaFold-derived model of LFRQ color coded by pLDDT score that highlights the confidence in each predicted region (blue being the highest confidence, red the lowest). (**C**) Functional domains of FRQ showing the start position for the LFRQ isoform (AUG$_{LFRQ}$.), key regions and the four native cysteine residues. Predicted secondary structure elements are given below as helices (zigzags) and β-strands (arrows). Color bar shows pLDDT scores across the sequence. Characterized domains include: CC, coiled coil domain; FCD, FRQ:CK1 interacting domain; PEST, proline, glutamic acid, serine, and threonine-rich domain; and FFD, FRQ:FRH interacting domain.

The online version of this article includes the following figure supplement(s) for figure 1:

**Figure supplement 1.** Structure prediction of FRQ.

**Figure supplement 2.** Domain map of FRQ with phosphosites (as identified by mass-spec) highlighted.

have been identified (*Figure 1C*), there is little information on the spatial arrangement of the FFC components.

Despite low-sequence conservation among the core negative arm proteins across species, they all share a high degree of disorder (*Pelham et al., 2020*). FRQ, the analogous *Drosophila* (d) Period (PER) protein, and the analogous mammalian (m) PER have >65% disorder predicted in their sequences with ~86% of the FRQ sequence assumed to have no defined structure (*Pelham et al., 2020*). Both FRQ and mPER2 have biochemical properties characteristic of intrinsically disordered proteins (IDPs) (*Hurley et al., 2013*; *Marzoll et al., 2022*). These clock repressors also undergo extensive post-translational modification (PTM), particularly phosphorylation (*Dunlap and Loros, 2018*; *Pelham et al., 2020*). FRQ acquires over 100 phosphorylation sites over the course of the circadian day (similar to PER phosphorylation by DBT), predominantly due to CK1 (*Baker et al., 2009*; *Tang et al., 2009*). The phosphorylation events have two consequences: (1) those that weaken the interaction between FRQ and the WCC or CK1 (*Cha et al., 2015*; *Liu et al., 2019*) and (2) those that direct FRQ down

the ubiquitin-proteasome pathway (*Larrondo et al., 2015*), although only the former are required for circadian rhythmicity (*Larrondo et al., 2015*). Multisite phosphorylation of FRQ influences its inhibitory function and clock period length (*Larrondo et al., 2015*; *Liu et al., 2019*). Although the consequences of phosphorylation-driven conformational changes in FRQ have been probed by proteolytic sensitivity, the challenges of producing FRQ have limited direct assessment of its conformational properties (*Liu et al., 2019*; *Pelham et al., 2020*; *Querfurth et al., 2011*). IDPs such as FRQ are known to participate in liquid–liquid phase separation (LLPS), which has emerged as a key mechanism for organizing cellular components, coordinating metabolism and directing signaling through, for example, the formation of membraneless organelles (MLOs) (*Dignon et al., 2020*; *Wright and Dyson, 2015*). Emerging studies in plants (*Jung et al., 2020*), flies (*Xiao et al., 2021*), and cyanobacteria (*Cohen et al., 2014*; *Pattanayak et al., 2020*) implicate LLPS in circadian clocks, and in *Neurospora* it has recently been shown that the Period-2 (PRD-2) RNA-binding protein influences *frq* mRNA localization through a mechanism potentially mediated by LLPS (*Bartholomai, 2022a*). Intriguing questions surround why clock repressor proteins are so disordered, undergo extensive PTM, and whether LLPS has a role to play in their function.

Clock proteins and their complexes have been challenging to characterize due to their large size (>100 kDa), low-sequence complexity, extensive disorder, and modification states. Herein, we reconstitute FRQ and the FFC to assess the global and site-specific structural properties of these moieties, as well as the dependence of their properties on multisite phosphorylation. FRQ undergoes non-uniform conformational change upon phosphorylation and arranges FRH close to CK1 in the FFC. We also demonstrate that FRQ indeed participates in phosphorylation-dependent LLPS in vitro, and as a condensate sequesters FRH and CK1 while altering CK1 activity. Our results suggest how a massively disordered protein such as FRQ organizes the clock core oscillator and reveal the effects of extensive phosphorylation on the conformational and phase-separation properties of a large IDP.

## Results

### Sequence assessment and general properties of FRQ

Analysis of an AlphaFold 2.0 (*Jumper et al., 2021*) model of full-length FRQ indicates that FRQ lacks a defined arrangement of large folded domains; instead, most of the molecule is unstructured (*Figure 1B*), in agreement with previous studies predicting that FRQ mostly comprises intrinsically disordered regions (*Hurley et al., 2013*). However, FRQ is also predicted to contain a small, clustered core, with several secondary structural elements that collapse together to form a compact nucleus surrounded by long flexible disordered regions (*Figure 1B*). The AlphaFold pLDDT scores, which represent the confidence in the structural prediction (*Jumper et al., 2021*), are very low across the model, except within the core (*Figure 1B*, *Figure 1—figure supplement 1*), which contains the CK1-binding region. The FRH-binding region, FFD, falls within a more flexible region with a lower pLDDT score than that of CK1. Helices compose the majority of secondary structural elements predicted by AlphaFold (*Figure 1B*). The assignments of the helical regions were well supported by the Agadir program that predicts helical propensity based on the helix/coil transition theory (*Muñoz and Serrano, 1994*). Four regions distributed throughout the length of FRQ (residues 194–212, 312–329, 709–719, and 801–817) have at least 10% helical propensity, and these helical regions mostly agree with those of the AlphaFold model (*Figure 1B*). Together, these analyses primarily indicate that FRQ assumes a broad conformational ensemble containing an ordered nucleus that interacts extensively with partially ordered and flexible regions. However, much of the structural details of the model, including the positioning and interactions between secondary structural elements, are predicted with low confidence, thereby underscoring the need for a multi-pronged biophysical study of the protein.

### Reconstitution of FRQ in unphosphorylated and highly phosphorylated states

The size and intrinsic disorder of FRQ have made it challenging to recombinantly express and purify, thereby limiting investigations of its biophysical properties. Previous purifications have characterized FRQ as an IDP, but suffered from low yields and aggregated species (*Hurley et al., 2013*; *Lauinger et al., 2014*). We also encountered extensive protein degradation and aggregation with conventional purification methods and affinity-tagging strategies; however, methods proven effective for IDPs

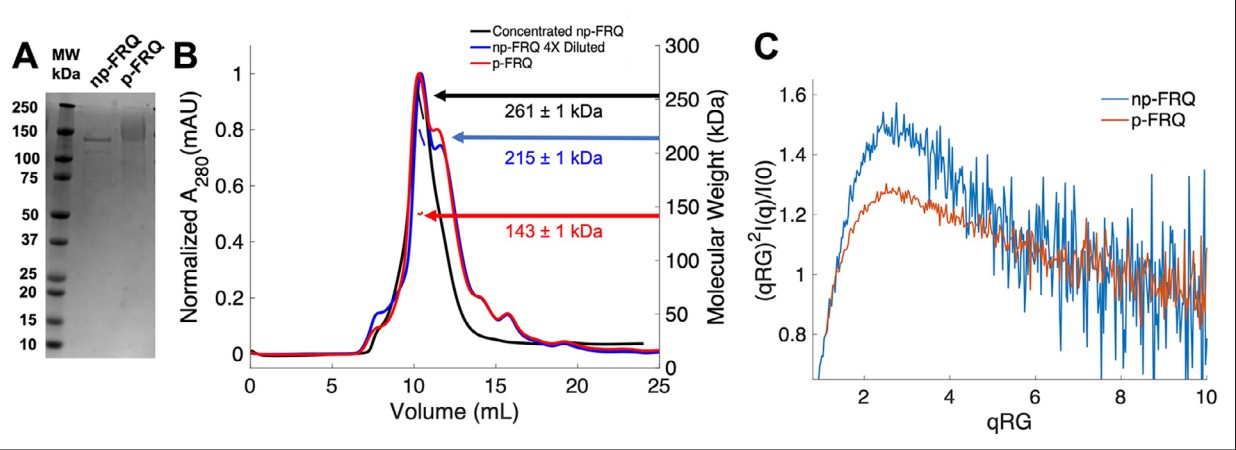

**Figure 2.** Purification of FRQ in non-phosphorylated and phosphorylated forms reveals intrinsically disordered protein (IDP) behavior. (**A**) SDS-PAGE gel showing non-phosphorylated FRQ (np-FRQ) and phosphorylated FRQ (p-FRQ) alongside a molecular weight ladder. Both samples represent the purified full-length long isoform (L-FRQ, residues 1–989) in non-denaturing buffer. (**B**) Size-exclusion chromatography coupled to multiangle light scattering (SEC-MALS) of phosphorylated FRQ (p-FRQ) and non-phosphorylated (np-FRQ). The molecular weight (MW) of p-FRQ (~143 ± 1 kDa) is somewhat larger than that of an unphosphorylated monomer (~120 kDa), but substantially smaller than that of a dimer, whereas the high-concentration MW value of np-FRQ (~261 ± 1 kDa) is slightly larger than that expected for a dimer (~240 kDa). Note that the p-FRQ MW is affected by >80 phosphosites. The shoulder to the right of the main peak had low light scattering and a large MW error. The colored arrows show the MW of each sample as determined by the MALS traces, which are inlaid within the SEC peaks. (**C**) Dimensionless Kratky plot of SEC-coupled small-angle X-ray scattering (SAXS) data for p-FRQ and np-FRQ; in both cases, the peak positions of the curves are shifted from that expected for a globular protein ($\sqrt{3}$,1.104), but neither plateau at 2, which would be characteristic of a flexible, denatured polypeptide (*Durand et al., 2010*). See *Table 1* for additional SAXS parameters. Total protein concentration was between 50 and 75 µM for the MALS and SAXS experiments.

The online version of this article includes the following figure supplement(s) for figure 2:

**Figure supplement 1.** Pairwise distance distributions calculated from size-exclusion chromatography-coupled small-angle x-ray scattering (SAXS) data for p-FRQ (red), np-FRQ (dark blue), p-FFC (magenta), and np-FFC (light blue).

(*Pedersen et al., 2020*) yielded comparatively well-behaved protein. Briefly, we carried out nickel affinity chromatography and size-exclusion chromatography (SEC) under denaturing conditions to maximize yield and reduce aggregation of full-length L-FRQ (residues 1–989; *Figure 2A*; *Seal et al., 2021*). Solubilization with high concentrations of guanidinium-HCl prevented proteolysis, aggregation, and limited co-purification of impurities that were otherwise difficult to remove. Once FRQ was isolated, the protein remained soluble, monodisperse, and could be concentrated after the denaturant was removed by dialysis (*Figure 2A and B*). To our knowledge, FRQ is one of the largest native IDPs produced in amounts amenable to structural and biophysical characterization.

To generate phosphorylated FRQ, the protein was co-expressed with (untagged) CK1, which forms a stable complex with FRQ and acts as its primary kinase (*Querfurth et al., 2011*). The same denaturing purification procedure was followed as for FRQ expressed without CK1. Phosphorylation was confirmed visually by altered migration on SDS-PAGE (*Figure 2A*) and mass spectrometry (*Figure 1—figure supplement 2*). The latter revealed a network of >80 phosphorylation sites (*Figure 1—figure supplement 2*), most of which agree with those found on natively expressed FRQ (*Baker et al., 2009*; *Tang et al., 2009*). Hereafter, we refer to this preparation as phosphorylated FRQ (p-FRQ), whereas non-phosphorylated FRQ (np-FRQ) refers to FRQ expressed without CK1. In general, we found that co-expression of FRQ with CK1 increased protein yield and stability, while limiting aggregation and degradation.

## Global properties of FRQ

Size-exclusion chromatography coupled to multiangle light scattering (SEC-MALS) of p-FRQ gives a molecular weight measurement consistent with a primarily monomeric state (molecular weight [MW] of ~143 kDa versus a predicted MW of ~110 kDa; *Figure 2B*). In contrast, the np-FRQ measured by MALS indicates a dimer and its measured mass distribution increases upon concentration (*Figure 2B*). Hence, purified np-FRQ dimerizes, whereas phosphorylation destabilizes this association. SEC-coupled small-angle X-ray scattering (SAXS) analyses revealed that both p-FRQ and np-FRQ shared

**Table 1.** SAXS parameters for p-FRQ, np-FRQ, and their reconstitution into the FFC.

| | $R_g$ [Guinier] (Å) | $R_g$ [P(r)] (Å) | Dmax (Å) | MW [$V_C$] (kDa) |
|---|---|---|---|---|
| p-FRQ | 123 ± 7 | 142 ± 5 | 530 ± 60 | 816 |
| np-FRQ | 100 ± 2 | 116 ± 4 | 446 ± 40 | 680 |
| p-FFC | 101 ± 2 | 136 ± 5 | 510 ± 20 | 314 |
| np-FFC | 8 0 ±2 | 109 ± 7 | 480 ± 10 | 286 |

Molecular weight calculations for p-FRQ and np-FRQ are inaccurate because $V_C$ (volume of correlation) plots (integrated area of I(q)q vs q; q = 2π/d) did not converge for these species. Expected MW for LFRQ is 110 kDa. Expected FFC MW for the 1:1:1 stoichiometry of FRQ:FRH:CK1 Is 273 kDa. Errors in Dmax were estimated from the range for which P(r) tailed to zero at high distances without constraint.

SAXS, size-exclusion chromatography-coupled small-angle X-ray scattering.

characteristics of highly flexible proteins, but that p-FRQ was more extended than np-FRQ (*Table 1*). The dimensionless Kratky plots of p-FRQ and np-FRQ, which report on compactness and conformational dispersity (*Durand et al., 2010*; *Martin et al., 2021Bateman et al., 2021*; *Sagar et al., 2020*; *Trewhella et al., 2017*), revealed high degrees of flexibility for both proteins (*Figure 2C*). The peak values of the plot were shifted from those expected for compact structures but were also not indicative of fully disordered proteins (*Durand et al., 2010*; *Rambo and Tainer, 2011*; *Sagar et al., 2020*). Furthermore, p-FRQ appears more disordered than np-FRQ, a characteristic also reflected in their pairwise distance distributions (*Figure 2—figure supplement 1*). These data then indicate that FRQ generally adopts a relatively monodisperse conformational state that is partially compact but highly flexible and then increases expanse and flexibility upon phosphorylation. This general picture is consistent with the increased susceptibility to proteolysis that FRQ experiences with phosphorylation (*Liu et al., 2019*; *Srivastava and Freed, 2019*; *Pelham et al., 2020*; *Querfurth et al., 2011*).

## Local measures of FRQ structure and dynamics

The properties of FRQ that include its intrinsic disorder and propensity to form heterogeneous ensembles are not well suited to commonly applied structural characterization methods. We applied site-specific spin labeling (*Figure 3A*) and electron spin resonance (ESR) spectroscopy to monitor FRQ local structural properties. FRQ has four native cysteine residues that provide reactive sites for nitroxide spin labels (SLs) (*Figure 3A and B*; *Berliner et al., 1982*). The Cys residues distribute relatively evenly throughout the protein sequence and fall within areas of varying structural order that include the binding regions for the core clock proteins (FRH, CK1, WC1, and WC2; *Tables 2 and 3*, *Figure 3B*). Furthermore, we developed and applied enzymatic SortaseA and AEP1 peptide-coupling methods (*Chandrasekaran et al., 2021*; *Nguyen et al., 2015*) to spin-label FRQ at its N- and C-termini (*Figure 3A*, *Figure 3—figure supplement 1*).

Line-shaped analysis from continuous wave (cw)-ESR spectroscopy reports on local dynamics (*Franck et al., 2015*; *Freed, 2005*). For rapidly rotating nitroxide SLs, the characteristic spectrum consists of three narrow lines of equal width and intensity. As the rotational mobility decreases, the lines broaden, especially the high-field line (*Bonucci et al., 2020*; *Emmanouilidis et al., 2021*). cw-ESR reveals that FRQ has regions of varying order that agree with computational predictions and that some of these sites undergo phosphorylation-dependent conformational changes (*Figure 3C*). SLs attached to Cys490 and Cys658 (490SL and 658SL) are predicted to be within an α-helix and a β-strand, respectively. These sites correspondingly showed a broad low-field (left-most line) component and a small high-field component (right-most line) indicative of slower tumbling compared to sites predicted to be in flexible regions, such as 746SL (*Figure 3C*). The 490SL site had a broader profile compared to that of the 658SL site, although both are relatively well ordered. We observed a similar trend with δ, which is the peak-to-peak width of the central line (*Hubbell et al., 2000*). 490SL and 658SL had similar δ values that were higher than that of 746SL, indicating that they are less flexible than 746SL (*Table 4*). Furthermore, the spectra also highlighted structural differences between the phospho forms: p-FRQ and np-FRQ gave very different δ values at 490SL and the C-terminus, thereby demonstrating how phosphorylation can influence conformational properties site specifically. The C-terminus may have become more dynamic in p-FRQ compared to np-FRQ because the C-terminus contains a cluster of phosphosites (*Figure 3C*). In contrast, the N-terminus and especially 490SL, which locates to the

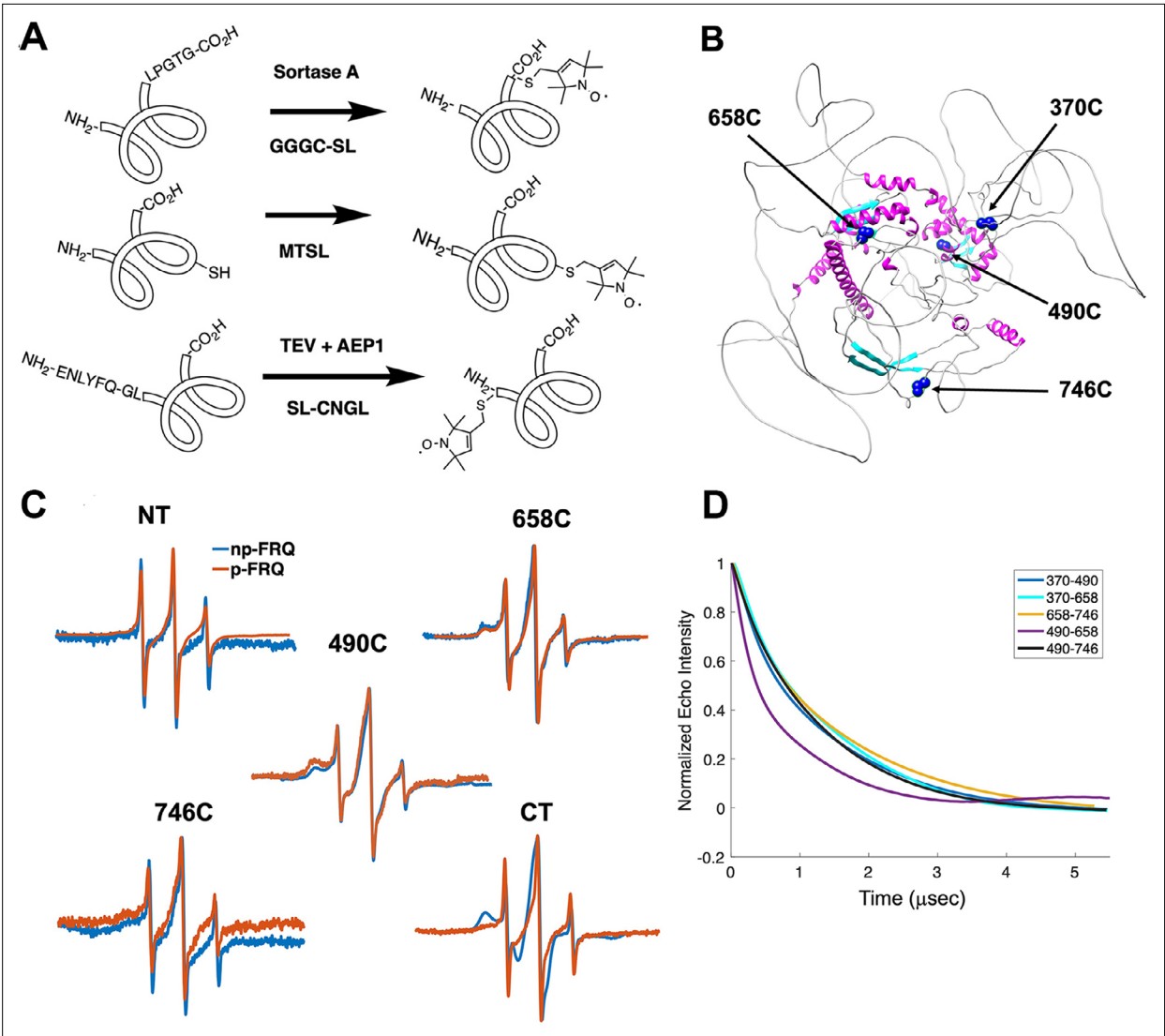

**Figure 3.** Electron spin resonance (ESR) spectroscopy studies of spin-labeled FRQ. (**A**) Spin-labeling strategies for FRQ. MTSL was used for internal labeling of cysteine residues and AEP1 and SortaseA were used to label the N and C terminus, respectively. TEV protease cleavage revealed the AEP1 ligation site. (**B**) AlphaFold-generated model of FRQ with the positions of native spin-labeled cysteine residues highlighted in dark blue. (**C**) X-band (9.8 GHz) room temperature continuous wave (cw)-ESR measurements for various spin-labeled FRQ variants, each containing a single Cys labeling site, as designated. Data represented as first-derivative spectra, with the horizontal axes depicting the static field strength (typically 3330–3400 G), and the vertical axis depicting the change in magnetic susceptibility with respect to the field, in arbitrary units. (**D**) Time-domain data of double electron–electron resonance (DEER) measurements for double-cysteine labeled FRQ variants. The baseline-corrected time-domain dipolar evolution trace fits are shown; raw data and baseline corrections are given in the supplements to this figure. The 490–658 separation is noticeably closer and more ordered than the others. The total protein concentration was between 50 and 75 µM for all experiments. The number of replicates varied for each spin pair, largely because there is considerable variation in spin-labeling efficiency. As FRQ protein samples are prone to behave poorly, they were generally produced until a well-labeled sample that remained soluble could be achieved. Nevertheless, the reproducibility of the data is reasonable. For example, P(r) data for p-FRQ 658–746C samples produced many months apart are identical within error.

The online version of this article includes the following figure supplement(s) for figure 3:

**Figure supplement 1.** Enzymatic labeling of FRQ at its termini.

**Figure supplement 2.** Time-domain traces before (raw) and after (data) background correction and trace fits (fit) from double electron–electron resonance (DEER) experiments of different double-cysteine FRQ variants alone or within the FFC (i.e., when FRQ was bound to FRH and CK1a).

**Table 2.** FRQ cysteine residue locations and context.

| Site | Domain | Binding partner at this region | Reference |
|------|--------|-------------------------------|-----------|
| 370C | N/A | | |
| 490C | FCD2 | CK1 | *Querfurth et al., 2011* |
| 658C | N/A | | |
| 746C | FFD | FRH | *Cheng et al., 2005* |

CK1-interaction region of the FCD$_2$ domain, are more ordered in p-FRQ. Such differences in mobility between p-FRQ and np-FRQ in the FCD (490SL) may reflect changes in FRQ:CK1 interactions as FRQ becomes progressively phosphorylated across the circadian period (*Liu et al., 2019*).

Double electron–electron resonance (DEER) spectroscopy provides distance distribution between two nitroxide labels in the range of 15 Å to ~100 Å. We produced six cysteine pair variants that each contained only two of the four native FRQ Cys residues to measure pairwise distances and one single Cys variant to test for oligomerization. Owing to association and aggregation when concentrated, np-FRQ was not amenable to DEER measurements; however, such issues were not prohibitive with p-FRQ. Nevertheless, due to the challenges of spin-labeling appreciable concentrations of p-FRQ, the modulation depths of the DEER data, which reflect the number of spins in range to produce specific dipolar interactions, are small (~2–4%). These values compare to ~10–12% for a well-ordered, highly labeled pair of sites under these instrument conditions (*Dunleavy et al., 2023*). The single Cys spin-labeled variant 490SL produced log-linear time-domain traces indicative of a uniform background with little evidence of dimerization, consistent with the MALS data showing p-FRQ to be monomeric (*Figure 3—figure supplement 2*). A second single Cys variant, 746C, likewise produced little dipolar signal, although for this variant, solubility issues limited spin concentration (*Figure 3—figure supplement 2*). The double 370SL:746SL p-FRQ variant also gave no dipolar signals and hence no dimerization as reported by those two sites (*Figure 3D*). When at least one of the two SLs resides outside of the predicted core, the DEER experiments of the spin pairs generally revealed small, albeit significantly slow decaying time-domain traces that deviate from background (*Figure 3D*, *Figure 3—figure supplement 2*, *Figure 4—figure supplement 1*). Such traces are indicative of long inter-spin distances and conformational flexibility, which would be consistent with the large mobility and intrinsic disorder of these peripheral regions (*Figure 3—figure supplement 2*, *Figure 4—figure supplement 1*). Only the variant 490SL:658SL gave a fast-decaying time-domain signal, indicative of short inter-spin distances and conformational rigidity (*Figure 3—figure supplement 2*, *Figure 4—figure supplement 1*). Despite the overall uncertainty in the AlphaFold model (*Figure 1B*), it is notable that 490SL and 658SL both fall within the predicted ordered core.

## FFC reconstitution and component arrangement

The FFC was generated by mixing p-FRQ or np-FRQ with excess FRH and CK1, followed by SEC purification to give p-FFC or np-FFC, respectively. FRHΔN (residues 100–1106), which binds FRQ, was used to improve expression and enhance stability (*Conrad et al., 2016*; *Hurley et al., 2013*). The resulting symmetric SEC traces and SDS-PAGE gels revealed pure and homogeneous ternary complexes in each case (*Figure 4A*). As expected, CK1 phosphorylates FRQ in np-FFC (*Figure 4A*, *Figure 4—figure supplement 2*), and this activity does not require priming by other kinases (*Marzoll et al., 2022*). SEC-SAXS analysis of the FFC showed that both FRQ phospho-forms produced complexes that were more globular than FRQ alone, but still somewhat flexible, with the p-FFC being more flexible than the np-FFC (*Figure 4B*, *Figure 2—figure supplement 1*). MW estimates of the p-FFC and

**Table 3.** FRQ regions that bind clock proteins.

| Binding partner | Region | Cysteine residue(s) present in this region | Reference |
|-----------------|--------|-------------------------------------------|-----------|
| FRH | 695–778 | 746C | *Cheng et al., 2005* |
| CK1 | 319–326; 488–495 | 490C | *Querfurth et al., 2011* |
| WC2 | | 490C, 658C, 746C | *Guo et al., 2010* |

**Table 4.** δ values from the cw-ESR central peak for FRQ variants of *Figure 3C*.

| Site | np-FRQ (G) | p-FRQ (G) |
|------|------------|-----------|
| NT | 1.75 | 1.87 |
| 490C | 1.97 | 2.47 |
| 658C | 1.97 | 2.01 |
| 746C | 1.76 | 1.82 |
| CT | 2.30 | 1.70 |

cw-ESR, continuous wave-electron spin resonance.

np-FFC from SAXS indicate a 1:1 stoichiometry of FRQ:FRH, with a likely single component of CK1 also contained within the complex (*Table 1*).

DEER measurements from selectively spin-labeled components within the complex were used to probe its overall architecture (*Figure 5*, *Figure 5—figure supplements 1–4*). An ADP-βSL probe (*Muok et al., 2018*) bound to both FRH and CK1 and revealed that the FRH and CK1 ATP-binding pockets are closely positioned within the FFC complex (*Figure 5A*, *Figure 5—figure supplement 1*). Nonetheless, the presence of broad and multiple peaks in the distance distributions indicated conformational heterogeneity. SLs placed within the FCD2 and FFD domains of FRQ (490C and 746C), which bind CK1 and FRH, respectively, produced a distance distribution similar to that of the labels that target the ATP-binding sites of the two enzymes (*Figure 5A*; *Cheng et al., 2005*). These independent observations support the conclusion that FRQ organizes FRH and CK1 in close proximity within the FFC. Notably, purified FRH and CK1 do not interact in the absence of FRQ (*Figure 5—figure supplement 1*).

SL separations within p-FRQ changed substantially when FRH and CK1 bound (*Figure 5*, *Figure 3—figure supplement 2*, *Figure 4—figure supplement 1*, *Figure 5—figure supplements 2–4*). In

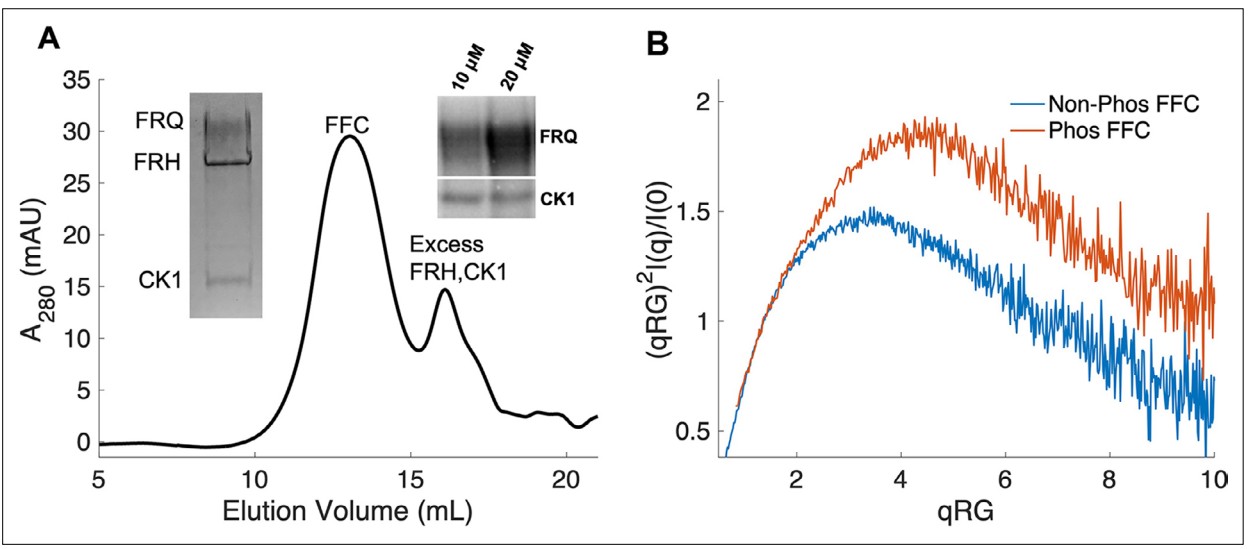

**Figure 4.** Properties of the FFC. (**A**) Size-exclusion chromatography (SEC) of the FFC; left inset: SDS-PAGE gel of the peak components; right inset: $^{32}$P-autoradiogram of FRQ phosphorylation by CK1 and CK1 autophosphorylation at 10 and 20 µM FRQ in the presence of FRH. Complete autoradiograms are shown in the supplements to this figure. (**B**) Overlay of dimensionless Kratky plots of SEC-coupled small-angle X-ray scattering (SAXS) data from the FFC formed with either np- or p-FRQ (protein concentrations were between 50 and 75 µM). The SAXS-derived molecular weight (MW) of np-FRQ FFC = 285 kDa, which compares to a predicted value of 273 kDa; MW of p-FFC = 340 kDa; *Table 1*.

The online version of this article includes the following figure supplement(s) for figure 4:

**Figure supplement 1.** Baseline-corrected time-domain fit traces from Q-band double electron–electron resonance (DEER) experiments of different double-cysteine FRQ mutants alone (blue) and within the FFC (orange).

**Figure supplement 2.** Complete autoradiograph of np-FRQ phosphorylation by CK1 and γ-$^{32}$P-ATP from *Figure 4A* inset.

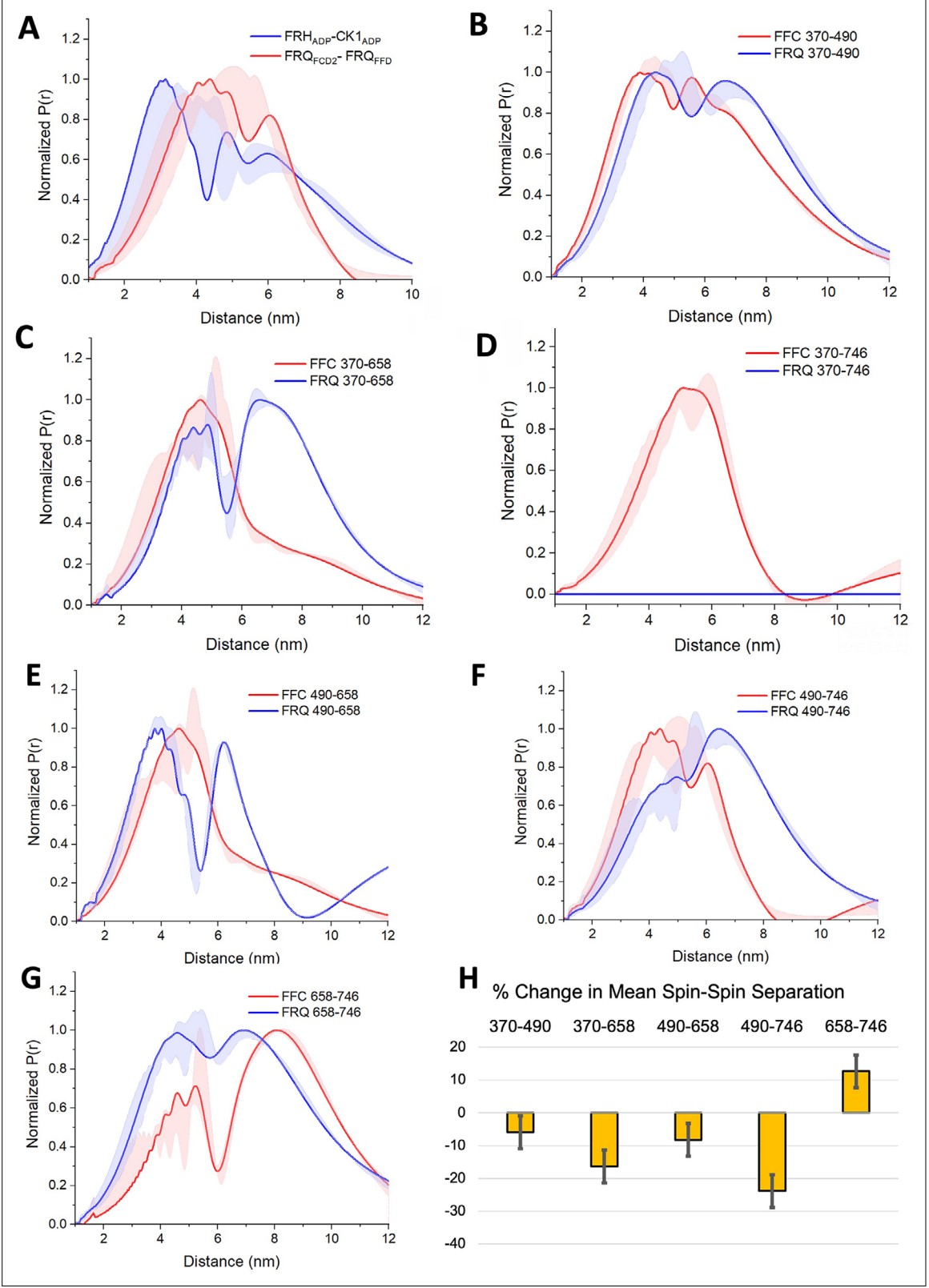

**Figure 5.** PDS (Pulsed dipolar ESR spectroscopy) distance distributions of spin-labeled FRQ and the FFC. (**A**) Double electron–electron resonance (DEER)-derived distance distributions of the FFC wherein the spins were directed by ADP linkage to target the FRH and CK1 ATP-binding sites (blue) or in the form of MTSL labeling of cysteine residues within the FCD2 and FFD domains of FRQ (red). Uncertainties in the distributions at each distance point (see Materials and Methods) are represented by the widths of the blue and red shading. Time-domain data and distance distributions with error

*Figure 5 continued on next page*

*Figure 5 continued*

estimates for single samples are shown in the supplements to this figure. (**B–G**) DEER-derived distance distributions of cysteine residue pairs labeled with MTSL within p-FRQ, either alone (blue) or within the FFC (red). Uncertainties in the distributions at each distance point are represented by the widths of the blue and red shading. Time-domain data and distance distributions with error estimates for single samples are shown in the supplements to this figure. (**H**) Percentage change in the average separation of FRQ when it binds FRH and CK1. Errors are derived from the uncertainties in the distance distributions (also shown in the supplements to this figure). In these experiments, only FRQ was labeled with MTSL at the positions noted. Note: total protein concentrations were between 50 and 75 μM for all experiments.

The online version of this article includes the following figure supplement(s) for figure 5:

**Figure supplement 1.** FRQ-mediated interaction between FRH and CK1.

**Figure supplement 2.** Error analysis of the distance distributions produced from double electron–electron resonance (DEER) experiments of different double-cysteine FRQ variants.

**Figure supplement 3.** Error analysis of the distance distributions produced from double electron–electron resonance (DEER) experiments of different double-cysteine FRQ variants within the FFC (i.e., when FRQ was bound to FRH and CK1).

**Figure supplement 4.** Gaussian function fitting of distance distributions.

general, the time-domain decays were more rapid in the FFC compared to p-FRQ alone, and the resulting distance distributions had smaller mean distances and less breadth (*Figure 5B–H*). However, the changes were not uniform; for example, the 658SL-to-746SL distance distribution increased in the complex compared to free FRQ (*Figure 5G*). To further assess how the distributions of p-FRQ conformations changed when the p-FFC formed, the distance data was fit to a model of one near and one far Gaussian functions (*Figure 5—figure supplement 4*, *Table 5*). Shifts in population between the near and far components were most prominent for the 370–658, 490–658, and 490–746 variants, which contain sites involved in core clock interactions (*Figure 5—figure supplement 4*). The proportion of the closer component increased from ~30% to ~70% in the 370–658 variant (*Figure 5—figure supplement 4*) and from ~10% to ~70% in the 490–746 variant (*Figure 5F*) when bound to FRH and CK1. Similar trends were observed when we compared the percentage change of the average separation observed in FRQ alone versus when it is bound to FRH and CK1 (*Figure 5H*). The 490, 658, and 746 sites reside in regions important for WCC interactions (*Guo et al., 2010*) and FRH may organize such binding determinants. Notably, the 370SL:746SL spin pair showed little dipolar coupling in free p-FRQ but resolved into an observable mid-range separation in the presence of CK1 and FRH (*Figure 5C*), perhaps because position 746 is within the FRH-binding region. Thus, CK1 and FRH order p-FRQ to an appreciable degree and the two enzymes are adjacently positioned within the conformationally heterogeneous complex.

## FRQ exhibits behavior consistent with LLPS in vitro and in vivo

The unstructured regions and multivalent interactions of IDPs can lead to LLPS (*Martin and Holehouse, 2020a*; *Martin et al., 2020b*). Such de-mixing often manifests in the formation of biomolecular condensates and MLOs within cells (*Posey et al., 2018*). As has been previously noted (*Pelham et al., 2020*), FRQ may be capable of LLPS owing to its extensive disorder. We found that FRQ has a high propensity to undergo LLPS based on a set of computational tools that evaluate charge distribution and clustering, long-range aromatic and non-aromatic π:π contacts, and nucleic acid-binding propensities (*Bolognesi et al., 2016*; *Vernon et al., 2018*). The overall FRQ PScore, which grades LLPS propensity on these properties (*Vernon et al., 2018*), was 6.39, with three regions

**Table 5.** R² values for the quality of the two-component Gaussian fits to the distance distributions.

|  | 370 | 490 | 658 | 746 |
|---|---|---|---|---|
| 370 |  | 0.99, 0.99 | 0.96, 0.99 | NA, 0.99 |
| 490 |  |  | 0.94, 0.99 | 0.99, 0.99 |
| 658 |  |  |  | 0.99, 0.95 |
| 746 |  |  |  |  |

There are two values reported in each cell – the first value corresponds to the R² value for FRQ-only distance while the second value is for the same variant in the FFC complex (i.e., when the FRQ is bound to CK1a and FRH).

scoring above a value of 2, the threshold for favorable LLPS (*Figure 6—figure supplement 1*). The region with the highest PScore also accounted for the majority of LLPS propensity as determined by the catGRANULE prediction algorithm (*Bolognesi et al., 2016*). Furthermore, we found that these methods also predicted the functional analogs dPER and hPER1 to phase separate (*Figure 6A*, *Figure 6—figure supplement 2*). The FRQ functional analogs share physicochemical characteristics with FRQ, particularly an enrichment of hydrophilic residues, increases in Gly and Pro content, and a depletion of hydrophobic residues, even though their positional sequence conservation with FRQ is low (*Figure 6—figure supplements 3 and 4*; see also *Jankowski et al., 2022*; *Pelham et al., 2021*). Similar amino acid composition patterns among FRQ, dPER, and hPER1 include the most common residues in each protein and its UniRef50 cluster (sequences with at least 50% identity with the target protein) being serine, proline, and glycine (~30% of total sequence) accompanied by a high depletion in aromatic residues (only ~7.5%) (*Figure 6—figure supplement 4*). In addition, all of the sequences have a reduced proportion of hydrophobic residues such as A, I, L, and V and a slightly acidic isoelectric point. Thus, FRQ, dPER, and hPER1 all have low overall hydropathy (*Figure 6—figure supplement 3*). However, in addition, FRQ and its homologs are unusual in their high Arg and Asp content, which tend to separate in charge blocks throughout their sequences (as indicated by high kappa values, *Figure 6—figure supplement 4*; see also *Jankowski et al., 2022*). Such charge clustering correlates with LLPS (*Somjee et al., 2020*). Overall, this analysis suggests that like intrinsic disorder, LLPS driven by the physiochemical properties of the constituent residues may be a conserved feature in the negative arm of the circadian clock.

We tested whether both np-FRQ and p-FRQ can undergo LLPS in vitro. In each case, upon exchanging the purified protein into their respective phase separation buffers (which lack any crowding agents), we observed the appearance of a turbid solution that contained microscopic droplets (*Figure 6B*). These droplets scaled in size and number with increasing protein concentration, dissolved upon the addition of 1,6-hexanediol, and were shown to dock and fuse to one another, thereby confirming their liquid-like properties (*Figure 6B*, *Figure 7—figure supplement 1*). UV-Vis turbidity assays as a function of temperature revealed that FRQ undergoes a low critical solution temperature (LCST) phase transition that is concentration and phosphorylation status-dependent (*Figure 6C*, *Figure 7—figure supplement 1*, *Tables 6 and 7*). As expected, phase separation increased as temperature and protein concentration increased as indicated by growing amplitudes and earlier phase-separation initiation temperatures (*Figure 6C*, *Figure 7—figure supplement 1*, *Tables 6 and 7*). Lower salt concentrations favored the LLPS of p-FRQ, whereas higher salt disfavored LLPS, compared to higher salt concentrations favoring LLPS for np-FRQ (*Figure 7—figure supplement 1*). Although the NaCl concentration for np-FRQ LLPS conditions is non-physiological per se, the ionic strength conditions and effective solute concentrations in cells are comparatively high (*Liu et al., 2017*). Furthermore, the higher absorbance values of p-FRQ at lower temperatures relative to that of np-FRQ indicate complex phase behavior for the phosphorylated isoform that cannot be adequately explained by an LCST phase transition alone (*Figure 7—figure supplement 1*). At temperatures below the approximated phase boundary indicated by the temperature transitions (*Figure 6C*, *Figure 7—figure supplement 1*), some p-FRQ phase separation had already occurred in the phase-separation buffer. Thus, FRQ phosphorylation alters its behavior consistent with LLPS as evidenced by its non-zero, concentration-dependent absorbance at lower temperatures, most likely by increasing the relative negative charges, changing the charge patterns, and providing more sites for multivalent interactions within the molecule (*Figure 6C*, *Figure 6—figure supplement 5*, *Figure 7—figure supplement 1*; *Somjee et al., 2020*; *Szabó et al., 2022*).

Live-cell imaging of *N. crassa* expressing FRQ tagged with the fluorescent protein mNeonGreen at its C-terminus (FRQ^mNeonGreen) revealed heterogeneous patterning of FRQ in nuclei (*Figure 6D and E*). FRQ thus tagged and driven by its own promoter is expressed at physiologically normal levels, and strains bearing FRQ^mNeonGreen as the only source of FRQ are robustly rhythmic with a slightly longer than normal period length. Live-cell imaging in *N. crassa* offers atypical challenges because the mycelia grow as syncytia, with continuous rapid nuclei motion during the time of imaging. This constant movement of nuclei is compounded by the very low intranuclear abundance of FRQ and the small size of fungal nuclei, making not readily feasible visualization of intranuclear droplet fission/fusion cycles or intranuclear fluorescent photobleaching recovery experiments (FRAP) that could report on liquid-like properties. Nonetheless, bright and dynamic foci-like spots were observed well inside the nucleus and

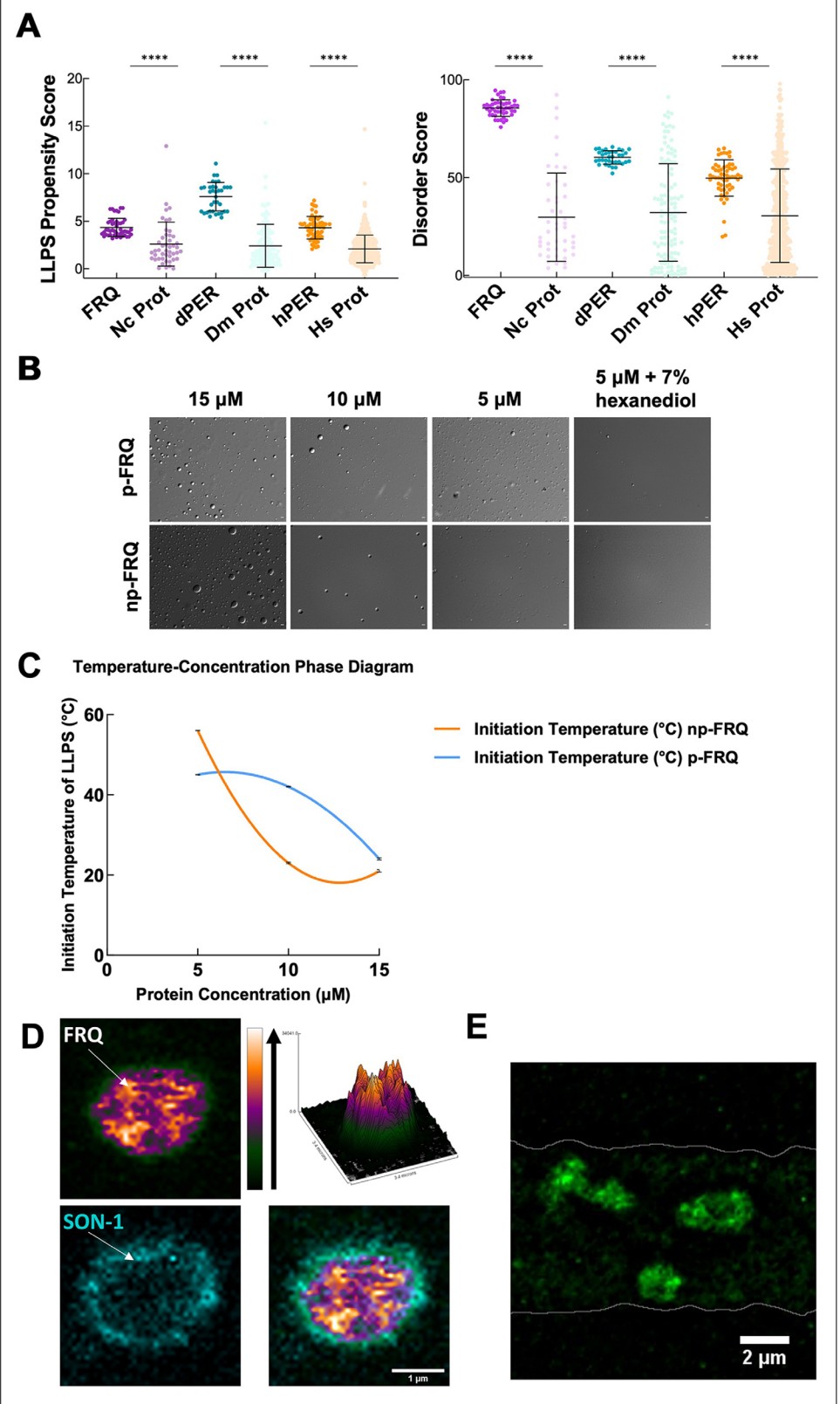

**Figure 6.** FRQ undergoes liquid–liquid phase separation (LLPS) in agreement with predictions from its sequence properties and those of its functional analogs. (**A**) LLPS propensity predictions using Pi-Pi and IUPred2A disorder predictions are shown for FRQ (purple), dPER1 (teal), and hPER1 (orange) compared to values for proteins of similar length from their respective proteomes (Prot): Nc, *Neurospora crassa*; Dc. *Drosophila melanogaster*;

*Figure 6 continued on next page*

*Figure 6 continued*

Hc, *Homo sapiens*. The mean and standard deviations are represented as horizontal and vertical black bars, respectively. Note: **** denotes a p-value<0.0001 obtained from a Mann–Whitney *U*-test. (**B**) Differential interference contrast (DIC) microscopy images of various concentrations of p-FRQ (in 150 mM NaCl, 25 mM Tris pH 8.0) and np-FRQ (in 500 mM NaCl, 25 mM Tris pH 8.0) at 25°C; images shown at ×100 magnification. Scale bar = 2 µm. (**C**) Temperature vs concentration phase diagram derived from the results of the turbidity assays shown in the supplements to this figure. Both p-FRQ and np-FRQ undergo an LCST phase transition on or above the line (a second-order polynomial fit to the data points). The error bars of each point (about the size of the points) reflect the 95% confidence interval of the mean. The phase boundary for p-FRQ represents the transition apparent in the temperature scans shown in the supplements to this figure. Some p-FRQ phase separation already occurs at low temperature under these buffer conditions, and thus, the region below the line does not represent fully soluble p-FRQ. (**D**) Live-cell fluorescent imaging of FRQ[mNeonGreen] in nuclei of *Neurospora* hyphae. Upper left panel shows the FRQ channel, represented as a multicolor heat map of fluorescence intensity; upper right panel shows a surface plot derived from the raw FRQ image to emphasize regions of concentration; lower left shows nucleoporin SON-1[mApple], which localizes to the cytoplasmic face of the nuclear envelope; bottom right shows the FRQ:SON-1 merge image. The images were acquired on a Zeiss 880 laser scanning confocal microscope and were smoothed by bicubic interpolation during tenfold enlargement from 42 × 42 pixels to 420 × 420 pixels. (**E**) FRQ[mNeonGreen] nuclear foci shown within an *N. crassa* syncytium mycelium (outlined by white lines), cropped from *Video 1*. Scale bar = 2 µm. Several hundred nuclei of identical genotype in multiple separate hyphae were examined. For visualization of intranuclear dynamics, 2–3 individual well-separated nuclei from multiple hyphae were examined in detail from each of 3 d. Representative data from one of these is reported.

The online version of this article includes the following figure supplement(s) for figure 6:

**Figure supplement 1.** FRQ, but not FRH or CK1, is highly disordered and predicted to undergo liquid–liquid phase separation (LLPS).

**Figure supplement 2.** FRQ and its functional homologs are predicted to undergo liquid–liquid phase separation (LLPS).

**Figure supplement 3.** FRQ and its functional homologs, dPER1 and hPER1, have similarly disparate physicochemical properties compared to their proteomes.

**Figure supplement 4.** FRQ and its functional homologs dPER1 and hPER1 have similar amino acid composition compared to their respective proteomes.

**Figure supplement 5.** Phosphorylation modulates the physicochemical properties of FRQ.

near the nuclear periphery, which is delineated by the cytoplasm-facing nucleoporin Son-1 tagged with mApple at its C-terminus (*Figure 6D and E*, *Video 1*). Such foci are characteristic of phase-separated IDPs (*Bartholomai et al., 2022b*; *Caragliano et al., 2022*; *Gonzalez et al., 2021*; *Tatavosian et al., 2019*) and share similar patterning to that seen for clock proteins in *Drosophila* (*Meyer et al., 2006*; *Xiao et al., 2021*), although the foci we observed are substantially more dynamic than those reported in *Drosophila*.

## Phase-separated FRQ recruits FRH and CK1 and alters CK1 activity

The PScore and catGRANULE LLPS predictors gave low overall scores for the FRQ partners, FRH and CK1 (*Figure 6—figure supplement 2*). This result is unsurprising because these proteins consist of

**Table 6.** Calculated values from the UV-Vis turbidity assays of np-FRQ shown in *Figure 6C*.

|  | 5 µM | 10 µM | 15 µM |
| --- | --- | --- | --- |
| Mean initial baseline absorbance | 0.02 | 0.08 | 0.13 |
| Temperature of initiation (°C) | 56 | 23 | 21 |
| Cloud point (inflection point) (°C) | 68 | 58 | 57 |
| Absorbance at inflection point | 0.07 | 0.21 | 0.33 |
| Slope at inflection point | 0.01 | 0.02 | 0.03 |
| Mean final baseline absorbance | 0.14 | 0.47 | 0.64 |

Initiation temperature was determined as baseline + 5× standard deviation of the baseline.

**Table 7.** Calculated values from the UV-Vis turbidity assays of p-FRQ shown in *Figure 6C*.

| | 5 µM | 10 µM | 15 µM |
|---|---|---|---|
| Mean initial baseline absorbance | 0.09 | 0.21 | 0.46 |
| Temperature of initiation (°C) | 45 | 42 | 24 |
| Cloud point (inflection point) (°C) | - | - | - |
| Absorbance at inflection point | - | - | - |
| Slope | 0.00 | 0.01 | 0.02 |
| Mean final baseline absorbance | - | - | - |

Initiation temperature was determined as baseline + 5× standard deviation of the baseline. p-FRQ continues to undergo liquid–liquid phase separation within the temperature range of the experiment and thus a cloud point cannot be calculated.

predominantly folded domains. Nonetheless, FRH and CK1 may still act as clients that localize to the FRQ-formed LLPS-like droplets. We chose to investigate whether FRH and CK1 could be recruited into droplets with fluorescent microscopy. Structured proteins are not well tolerated in the temperature-dependent turbidity assays because of the large signal that they give upon domain unfolding. We found that fluorescently labeled FRH and CK1 were recruited into phase-separated FRQ droplets, but a control protein labeled with the same fluorophore, *Escherichia coli* CheY, was not (*Figure 7A*). The ability of FRQ to recruit its partner proteins into phase-separated droplets provides a means to sequester them and potentially control their availabilities and activities.

LLPS has been suggested to modulate the activities of enzymes by creating a microenvironment to alter substrate sequestration and product release (*Shapiro et al., 2021*). We tested the effect of FRQ LLPS on its phosphorylation by CK1, a key reaction for regulating the period of the clock (*Liu et al., 2019*). We compared the phosphorylation of FRQ by CK1 in a buffer that supports phase separation under different temperatures using the latter as a means to control the degree of LLPS without altering the solution composition. Both the UV-Vis turbidity assays and the differential interference contrast (DIC) micrographs of the protein at different temperatures establish that FRQ LLPS increases substantially with temperature (*Figures 6C* and *7C*). Furthermore, LLPS of p-FRQ and np-FRQ both depend highly on buffer composition (*Figure 7—figure supplement 1*). CK1 autophosphorylation served as an internal control and its temperature dependence was consistent with previous studies that show increased catalytic activity at higher temperatures (*Marzoll et al., 2022*). As temperature increases drove more LLPS, the phosphorylation of FRQ decreased, even as CK1 autophosphorylation increased (*Figures 7B* and *8*). Quantitatively, we observed a reduction in FRQ phosphorylation by twofold under LLPS conditions relative to conditions under which it does not phase separate (*Figure 7B*).

Through application of cw-ESR, we probed FRQ dynamics when it undergoes LLPS. We observed a striking rigidification of spin-labeled FRQ when LLPS was promoted by buffer conditions (*Figure 7D*). The reduction of CK1-mediated FRQ phosphorylation, which likely requires substantial dynamics given the distribution of phosphorylation sites throughout the FRQ sequence, may reflect this change in molecular dynamics. Thus, in addition to affecting the localization of FRQ and its partners, LLPS also impacts the enzymatic activity of the complex by perhaps modulating molecular encounters and motions essential for large-scale phosphorylation.

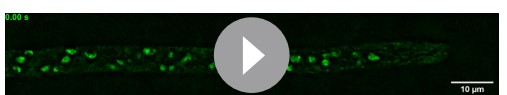

**Video 1.** Visualization of FRQ[mNeonGreen] in living *Neurospora*, displaying heterogeneous, punctate intranuclear localization. Images of a single focal plane were acquired in a time series with 300 ms between exposures.

https://elifesciences.org/articles/90259/figures#video1

## Discussion

Within the TTFLs that define eukaryotic circadian clocks, large repressor proteins like FRQ compose cellular timing mechanisms by coordinating transient protein–protein interactions, nuclear entry, targeted protein degradation, and transcriptional repression (*Pelham et al., 2020*). They compose dynamic macromolecular assemblies that achieve specificity in their functions despite assuming highly variable structures (*Pelham et al., 2020*).

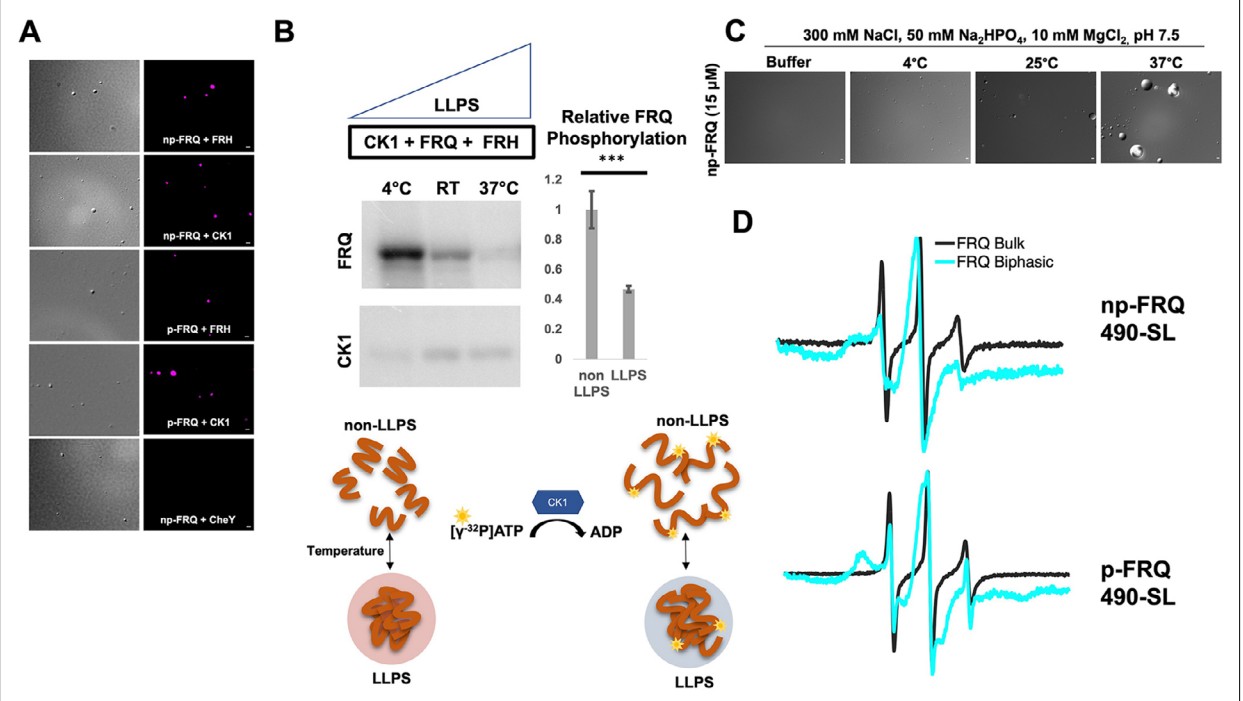

**Figure 7.** Conditions favoring liquid–liquid phase separation (LLPS) alter the structural and enzymatic properties of the FFC. (**A**) Differential interference contrast (DIC) and fluorescence microscopy images at 25°C of phase-separated np-FRQ or p-FRQ (5 µM each) droplets in 500 mM KCl, 20 mM Na₂HPO₄, pH 7.4 with either equimolar Cy5-labeled FRH, CK1, or CheY (control) visualized using Cy5 fluorescence. Scale bar = 2 µm. (**B**) Quantification of FRQ phosphorylation by CK1 at RT under LLPS (n=2) and non-LLPS (n=4) conditions with phosphorylation levels under non-LLPS conditions normalized to 1 (Top) Autoradiography of FRQ (20 µM) (left) phosphorylated by CK1 in the presence of FRH at increasing temperatures. (Bottom) Schematic showing the nature and results of the autoradiography assay depicted above. The FRQ phosphorylation under LLPS conditions was reduced relative to non-LLPS conditions. The complete autoradiograms are shown in the supplements to this figure. Quantification of FRQ phosphorylation by CK1 at RT under LLPS and non-LLPS conditions with phosphorylation levels under non-LLPS conditions normalized to 1. Note: *** denotes a p-value<<0.05 obtained from a Student's *t*-test. (**C**) DIC microscopy images of 15 µM phase-separated np-FRQ under the same conditions (i.e., buffer and temperature) as the phosphorylation assay shown in (**B**). Scale bar = 2 µm. (**D**) X- band (9.8 GHz) RT continuous wave-electron spin resonance (cw-ESR) spectra of FRQ labeled with MTSL at the 490 site in solubilizing buffer (black) and under conditions that promote LLPS (cyan). Data represented as first-derivative spectra, with the horizontal axes depicting the static field strength (typically 3330–3400 G), and the vertical axis depicting the change in magnetic susceptibility with respect to the field, in arbitrary units.

The online version of this article includes the following figure supplement(s) for figure 7:

**Figure supplement 1.** Exploring the phase behavior of FRQ.

**Figure supplement 2.** Complete autoradiograph of np-FRQ phosphorylation by CK1 and γ-³²P-ATP.

Our purification scheme enabled a multipronged biophysical characterization of FRQ. The majority of previous such studies on IDPs, particularly those that undergo de-mixing, have involved truncated proteins, excised domains, or peptides of sizes less than 200 residues (*Babinchak et al., 2019*; *Emmanouilidis et al., 2021*; *Lin et al., 2021*; *Lin et al., 2019*; *Seal et al., 2021*). FRQ represents a full-length, nearly 1000-residue IDP that can be investigated with respect to PTM, interaction partners, and phase separation. As anticipated from its sequence (*Hurley et al., 2013*; *Pelham et al., 2020*), this direct characterization of structural properties shows that FRQ is indeed a highly dynamic protein, but even in isolation, it is not completely disordered. The SAXS and ESR data indicate that FRQ has compact features that localize to an ordered core interspersed with dynamic regions. Notably the shape of the Kratky plots generated from the SAXS data suggests a degree of disorder that is substantially greater than that expected of a molten globule (*Kataoka et al., 1997*), but far from that of a completely denatured protein (*Kikhney and Svergun, 2015*; *Martin et al., 2021Bateman et al., 2021*). Similarly, the DEER distributions, though non-uniform across the various sites examined, indicate more disorder than that of a molten globule (*Selmke et al., 2018*) but more order than a completely unfolded protein (*van Son et al., 2015*). Although FRQ lacks the well-defined PAS domains of the analogous PER proteins (*Pelham et al., 2020*), it also maintains a structured core to

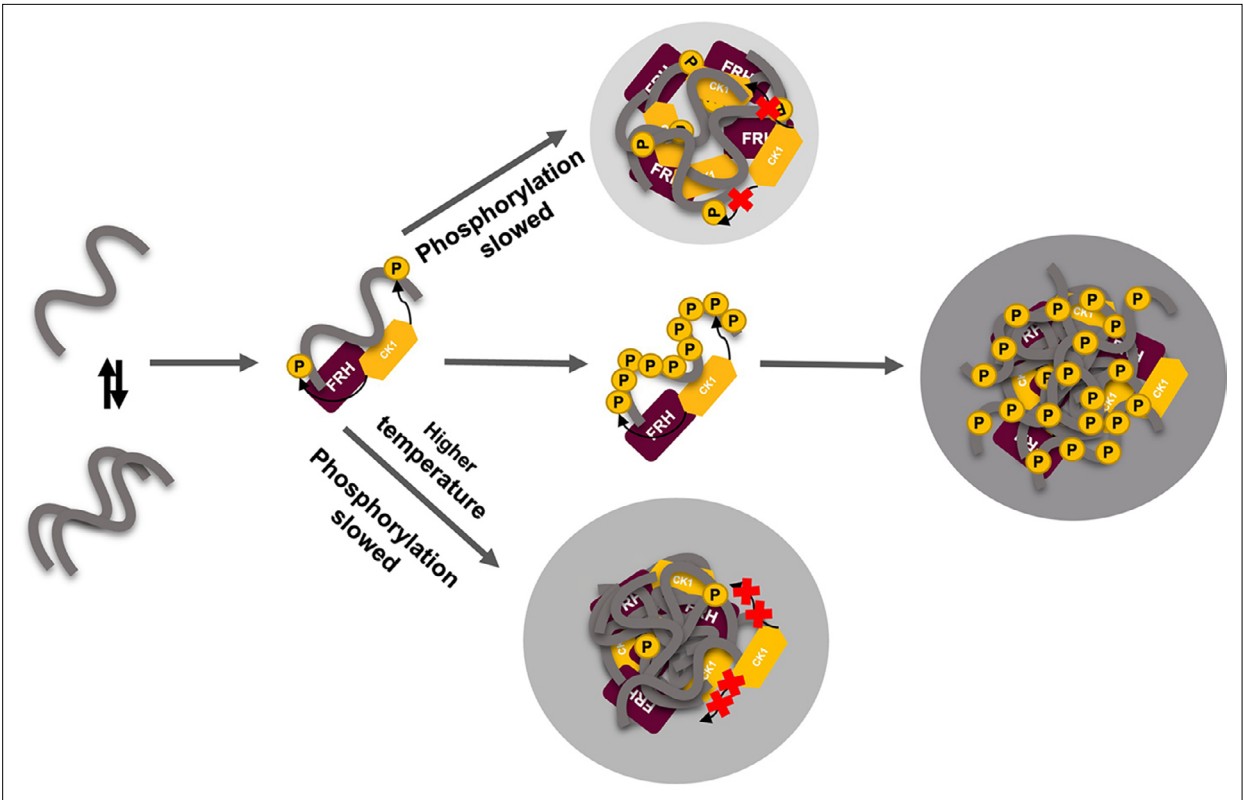

**Figure 8.** Liquid–liquid phase separation (LLPS) may play a role in temperature compensation of the clock through modulation of FRQ phosphorylation. Reduced CK1 phosphorylation of FRQ causes both longer periods (*Mehra et al., 2009*) and loss of temperature compensation (manifested as a shortening of period at higher temperature) (*Hu et al., 2021*; *Liu et al., 2019*). Thus, the ability of increased LLPS at elevated temperature (larger gray circle) to reduce FRQ phosphorylation by CK1 will counter a shortening period that would otherwise manifest in an undercompensated system. As further negative feedback, LLPS is also promoted by increased FRQ phosphorylation, which in turn will reduce phosphorylation by CK1. Thus, both increased FRQ phosphorylation and temperature favor LLPS and reduction of CK1 activity.

anchor its more dynamic elements. The regions that bind CK1 and FRQ are the most ordered, and they organize the two partners beside each other; in fact, they are in remarkably close proximity, as judged by SLs in their ATP-binding pockets or in their binding domains. FRH mediates contacts of the FFC to the WCC (*Conrad et al., 2016*), whereas CK1 phosphorylates the WCC to close the TTFL (*Cha et al., 2015*; *Larrondo et al., 2015*; *Liu et al., 2019*; *Wang et al., 2019*); hence, the targeting of CK1 to the WCC likely depends on FRQ associating FRH with CK1 (*Wang et al., 2019*).

Previous proteolytic sensitivity studies have suggested that increased phosphorylation across the day transitions FRQ from a 'closed' to a more 'open' state (*Liu et al., 2019*; *Pelham et al., 2020*; *Querfurth et al., 2011*). Our data paint a more nuanced picture. The SAXS data reveal that p-FRQ and p-FFC are more expansive and show greater flexibility than do np-FRQ and np-FFC, respectively. Furthermore, np-FRQ has a greater tendency to self-associate. In the cell, completely unphosphorylated FRQ is likely short-lived, if present at all (*Baker et al., 2009*). Phosphorylation may stabilize FRQ against aggregation, yet also provides for enough conformational heterogeneity to poise FRQ near a degradation threshold (*Larrondo et al., 2015*). This idea is supported by the cw-ESR data (*Figure 3*), which shows that phosphorylation causes non-uniform local structural changes with some regions increasing, while others decreasing, in flexibility. The DEER data also demonstrates that binding to FRH and CK1 increases compactness of p-FRQ, particularly in the central core that is responsible for binding the partners. In contrast, the FRQ C-terminus gains considerable flexibility in the FFC. FRQ has been suggested to assume a dimeric state because of a predicted coiled-coil region and formation of mixed dimers formed in cellular pull-down assays with differentially tagged FRQ subunits (*Cheng et al., 2001*). Np-FRQ does indeed dimerize when isolated, whereas p-FRQ is primarily monomeric alone. Furthermore, SAXS data indicates that there is one copy of FRQ in both the reconstituted np-FFC and p-FFC. Hence, phosphorylation has the potential to regulate the FRQ

oligomeric state, which may in turn be linked to FFC assembly. Despite being a monomer in isolation, the pull-down data (*Cheng et al., 2001*) indicate that association of p-FRQ may be mediated by other cellular factors, including LLPS.

FRQ has physicochemical features embedded within its sequence that give rise to LLPS and the FRQ clock analogs exhibit similar characteristics (*Figure 6—figure supplements 2–4*). FRQ and negative-arm proteins in general belong to the 'flexible disorder' class of IDPs (*Shapiro et al., 2021*), wherein amino acid patterns that generate intrinsic disorder are conserved, but amino acid sequence per se is not (*Bellay et al., 2011*). Moreover, although these proteins all clearly contain regions of low-complexity sequence (*Hurley et al., 2013*; *Marzoll et al., 2022*; *Pelham et al., 2020*), their conserved features include not only their intrinsic disorder, but also their propensity for LLPS (*Figure 6A*, *Figure 6—figure supplement 2*). These two properties, although related, are not directly correlated (*Bolognesi et al., 2016*; *Martin et al., 2020b*; *Szabó et al., 2022*; *Vernon et al., 2018*). In particular, a preponderance of aromatic and non-aromatic π-π planar contacts and charge-dense or charge-repeat regions drive LLPS (*Szabó et al., 2022*; *Vernon et al., 2018*) and these features are consistently found in clock repressors, particularly FRQ. The similar amino acid compositions of these proteins may then potentially give rise to their shared function. Moreover, extensive phosphorylation may alter LLPS propensity because it increases extended states, solvation of the peptide backbone, charge density, and intermolecular hydrogen bonding capacity (*Bolognesi et al., 2016*; *Szabó et al., 2022*; *Vernon et al., 2018*). For FRQ, the sequence predictions are borne out experimentally, with p-FRQ displaying a different phase behavior relative to np-FRQ. In addition to being more soluble, dynamic, and less aggregated than np-FRQ, p-FRQ also phase separates at lower temperatures and under lower salt conditions relative to np-FRQ. Increased salt disfavors p-FRQ phase separation, which, when compared to np-FRQ, is consistent with more electrostatically driven de-mixing due to the high degree of phosphorylation.

Live-cell imaging of fluorescently tagged FRQ proteins is consistent with FRQ phase separation in *N. crassa* nuclei. FRQ is plainly not homogeneously dispersed within nuclei, and the concentrated foci observed at specific positions in the nuclei indicate condensate behavior similar to that observed for other phase-separating proteins (*Bartholomai, 2022a*; *Caragliano et al., 2022*; *Gonzalez et al., 2021*; *Tatavosian et al., 2019*; *Xiao et al., 2021*). While ongoing experiments are exploring more deeply the spatiotemporal dynamics of FRQ condensates in nuclei, the small size of fungal nuclei as well as their rapid movement with cytoplasmic bulk flow through the hyphal syncytium makes these experiments difficult. Of particular interest is drawing comparisons between FRQ and the *Drosophila* Period protein, which has been observed in similar foci that change in size and subnuclear localization throughout the circadian cycle (*Meyer et al., 2006*; *Xiao et al., 2021*), although it must be noted that the foci we observed are considerably more dynamic in size and shape than those reported for PER in *Drosophila* (*Xiao et al., 2021*). A very recent manuscript (*Xie et al., 2023*) calls into question the importance and very existence of LLPS of clock proteins at least in regards to mammalian cells, noting that it may be an artifact of overexpression in some instances where it is seen, and that at normal levels of expression there is no evidence for elevated levels at the nuclear periphery. Artifacts resulting from overexpression are unlikely to be a problem for our study and that of Xiao et al. as in both cases clock proteins were tagged at their endogenous locus and expressed from their native promoters. Although we noted enrichment of FRQ[mNeonGreen] near the nuclear envelope in our live-cell imaging, there remained abundant FRQ within the core of the nucleus.

Phosphorylation is known to modulate LLPS in other systems, but the effects are variable. For example, phosphorylation of the transcriptional silencing factor HP1α also enhances LLPS, whereas only the non-phosphorylated form of *Xenopus* CPEB4 involved in translation of mRNA poly-A tails undergoes LLPS (*Larson et al., 2017*; *Seal et al., 2021*). FRQ, especially p-FRQ, may achieve its inhibitory function by sequestering the FFC and WCC into MLOs. The large, disordered domains and capacity for multivalency of the WC proteins (*Pelham et al., 2020*) may also facilitate their partitioning into such condensates. In the fly clock, PER and the WCC analog CLK produce liquid-like foci in the nuclei that are most prevalent during the repressive phase of the circadian cycle (*Xiao et al., 2021*).

These large foci, which are few and localized to the nuclear periphery, may serve to organize clock-controlled genes during repression (*Xiao et al., 2021*). LLPS may also influence the nuclear transport of FRQ by affecting interactions with the nuclear import machinery as has been observed with proteins such as TD43 and FUS (*Gasset-Rosa et al., 2019*; *Gonzalez et al., 2021*). Conditions favoring LLPS reduce the ability of FRQ to act as a substrate for CK1 (*Figure 7B*), and this reactivity correlates with reduced dynamics of FRQ in the phase-separated state (*Figure 7D*). Domains and disordered regions from other spin-labeled proteins show similar reduced dynamics when induced to phase separate and monitored by cw-ESR spectroscopy (*Babinchak et al., 2019*; *Emmanouilidis et al., 2021*; *Lin et al., 2021*; *Seal et al., 2021*), although the temperature and salt dependencies of the ordering behavior can vary considerably, with some species showing little change upon LLPS (*Lin et al., 2019*).

In the cell, the impact of LLPS on CK1 activity may modulate the timing of FRQ phosphorylation, which is closely linked to clock period (*Larrondo et al., 2015*; *Marzoll et al., 2022*). Moreover, and collectively, the data from *Figures 6 and 7* show that as temperature increases, LLPS-like behavior increases, and as LLPS increases, the ability of CK1 to phosphorylate FRQ decreases. We believe that the reduced CK1 kinase activity toward FRQ as a substrate is directly due to the impact of the generated LLPS milieu, that is, the changes in structural/dynamic properties of FRQ and/or CK1 induced by the effects of being a phase-separated microenvironment, which could be substantially different from the non-phase-separated buffer environment. For example, previous work done on the disordered region of DDX4 (*Brady et al., 2017*; *Nott et al., 2015*) shows that even the amount of water content and stability of biomolecules such as double-strand nucleic acids encapsulated within the droplets differ between non- and phase-separated DDX4 samples. Indeed, the spin-labeling experiments indicate that the dynamics of FRQ have been altered by LLPS (*Figure 7D*).

The rate of FRQ phosphorylation dictates period length (*Larrondo et al., 2015*). Therefore, this negative feedback of temperature on period mediated by LLPS formation could provide a mechanism for circadian temperature compensation of period length (*Figure 8*). Sequestration of CK1 (and FRH) into condensates may also render these factors unavailable to target other cellular substrates. Phosphorylation alters FRQ interactions throughout the circadian cycle, not unlike the workings of the cyanobacterial clock, which relies upon a timed phospho-code to regulate enzymatic components (*Rust et al., 2007*; *Wang et al., 2019*). We demonstrate that FRQ likewise responds to phosphorylation with non-uniform conformational changes across its sequence and an increased propensity for LLPS. It is the latter property that appears conserved among negative-arm repressor proteins in eukaryotic clocks and may thus be a critical function of their intrinsic disorder and extensive PTMs.

## Materials and methods

**Key resources table**

| Reagent type (species) or resource | Designation | Source or reference | Identifiers | Additional information |
|---|---|---|---|---|
| Gene (*N. crassa*) | LFRQ S513A | Uniprot | Uniprot ID:P19970 | |
| Gene (*N. crassa*) | FRH | Uniprot | Uniprot ID:Q1K502 | |
| Gene (*N. crassa*) | CK1 | Uniprot | Uniprot ID:V5IR38 | |
| Strain, strain background (*Escherichia coli*) | BL21(DE3) | NEB | Catalog #: C2527H | Chemically competent cells |
| Genetic reagent (*N. crassa*) | frq[mNeonGreen-FLAG] | *Bartholomai, 2022a* | FungiDB ID: NCU02265 | |
| Genetic reagent (*N. crassa*) | wc-1[ΔLOV] | *Bartholomai, 2022a* | FungiDB ID: NCU02356 | |
| Genetic reagent (*N. crassa*) | son-1[mApple] | *Wang et al., 2023*; https://doi.org/10.1016/j.fgb.2022.103763 | FungiDB ID: NCU04288 | |
| Genetic reagent (*N. crassa*) | Δmus-51 | *Colot et al., 2006*; https://doi.org/10.1073/pnas.0601456103 | FungiDB ID: NCU08290 | |

*Continued on next page*

Continued

| Reagent type (species) or resource | Designation | Source or reference | Identifiers | Additional information |
|---|---|---|---|---|
| Cell line (include species here) | *Neurospora crassa* frq[mNeonGreen-FLAG] hph+, wc-1[ΔLOV] bar+, csr-1::pNCU04502-NCU04288(son-1)[mApple], Δmus-51::bar | **Bartholomai, 2022a** | | frq and wc-1 modified by targeted insertion at their resident loci using hph or bar respectively as selectable markers, son-1[mApple] inserted by targeted transformation to csr-1, mus-51 replaced by bar |
| Transfected construct (include species here) | *Neurospora crassa* frq[mNeonGreen-FLAG] hph+, wc-1[ΔLOV] bar+, csr-1::pNCU04502-NCU04288(son-1)[mApple], Δmus-51::bar | **Bartholomai, 2022a** | | frq and wc-1 modified by targeted insertion at their resident loci using hph or bar respectively as selectable markers, son-1[mApple] inserted by targeted transformation to csr-1, mus-51 replaced by bar |
| Biological sample (include species here) | | | | |
| Antibody | (Include host species and clonality) | | | (include dilution) |
| Recombinant DNA reagent | pET-28a | Novagen | Catalog #: 69864–3 | |
| Recombinant DNA reagent | LFRQ_ 370 S | This paper | | pET28 a backbone with twin strep tag Primers- GCGCGAAGCTTCAAT CCAACTACAG CTGTAGTTGGATTGAA GCTTCGCGC |
| Recombinant DNA reagent | LFRQ_ 490 S | This paper | | pET28 a backbone with twin strep tag Primers- CAATCTGCTTTCAAA CCTGGCCCAG CTGGGCCAGGTTTG AAAGCAGATTG |
| Recombinant DNA reagent | LFRQ_658 S | This paper | | pET28 a backbone twin strep tag Primers- CGCTCCATTTTCAAC CGATCTTTC GAAAGATCGG TTGAAAATGGAGCG |
| Recombinant DNA reagent | LFRQ_746 S | This paper | | pET28 a backbone with twin strep tag Primers- GTCTTCCCATGGAGT GAGGATCCTGC GCAGGATCCTCA CTCCATGGGAAGAC |
| Recombinant DNA reagent | LFRQ_LPGTG | This paper | | pET28 a backbone with twin strep tag Primers- CGATGGAGGACGTCT CATCCTCGCTGCCGGGCACCGGCT TTAAGCATTATG CGGCCGCTCAGCC GGTGCCCGGCAG |
| Recombinant DNA reagent | ENLYFQGL_LFRQ | This paper | | pET28 a backbone Primers- CCTGTACTTCCAAGGACTA GGAATGGCCGATAG CTATCGGCCATTCCTAGTCC TTGGAAGTACAGG |
| Sequence-based reagent | | | | |
| Peptide, recombinant protein | BSA | Sigma | CAS #: 9048-46-8 | |
| Peptide, recombinant protein | GGGGC | **Chandrasekaran et al., 2021** https://doi.org/10.1038/s42003-021-01766-2 | | |
| Peptide, recombinant protein | CNGL | **Nguyen et al., 2015** https://doi.org/10.1002/ange.201506810 | | |
| Peptide, recombinant protein | SortaseA | **Chandrasekaran et al., 2021** https://doi.org/10.1038/s42003-021-01766-2 | | |
| Peptide, recombinant protein | AEP1 | **Nguyen et al., 2015** https://doi.org/10.1002/ange.201506810 | | |

*Continued on next page*

*Continued*

| Reagent type (species) or resource | Designation | Source or reference | Identifiers | Additional information |
|---|---|---|---|---|
| Commercial assay or kit | | | | |
| Chemical compound, drug | MTSL | Cayman Chemicals | CAS #: 81213-52-7 | |
| Chemical compound, drug | ADP-β-S-SL | *Muok et al., 2018* https://doi.org/10.1007/s00723-018-1070-6 | | |
| Chemical compound, drug | Cy5-Maleimide | Cytiva | PA25031 | |
| Chemical compound, drug | [γ–32P] ATP | PerkinElmer | BLU002A001MC | |
| Software, algorithm | ASTRA IV | Wyatt Technology | | |
| Software, algorithm | RAW | *Hopkins et al., 2017* https://doi.org/10.1107/S1600576717011438 | | |
| Software, algorithm | GraphPad | Dotmatics | | |
| Software, algorithm | Fiji | *Schindelin et al., 2012*; doi:10.1038/nmeth.2019 | | |
| Software, algorithm | Matlab | MathWorks | | |
| Software, algorithm | localCIDER | *Holehouse et al., 2017* https://doi.org/10.1016/j.bpj.2016.11.3200 | | |
| Software, algorithm | IUPred2A | *Mészáros et al., 2018* https://doi.org/10.1093/nar/gky384 | | |
| Software, algorithm | Agadir | *Muñoz and Serrano, 1994* https://doi.org/10.1038/nsb0694-399 | | |
| Software, algorithm | AlphaFold2.0 | *Jumper et al., 2021* https://doi.org/10.1038/s41586-021-03819-2 | | |
| Software, algorithm | PScore | *Vernon et al., 2018*; https://doi.org/10.7554/eLife.31486 | | |
| Software, algoritm | catGRANULE | *Bolognesi et al., 2016* https://doi.org/10.1016/j.celrep.2016.05.076 | | |
| Other | | | | |

## Protein expression and purification

Full-length L-FRQ (UniProt ID:P19970), CK1 (CK1a; UniProt ID:V5IR38), and FRHΔN (residues 100–1106) (UniProt ID:Q1K502) were sub-cloned into a pET28a vector (Novagen) bearing an N-terminal His$_6$-tag and kanamycin resistance. Note that L-FRQ contained the S513A substitution to improve protein stability as noted in *Querfurth et al., 2007*. Cysteine to serine substitution mutants of FRQ were produced using standard mutagenesis, with the primers described in the Key Resources Table. The plasmids encoding the proteins were transformed into BL21 (DE3) *E. coli* cells, grown in LB media containing 50 µg/mL kanamycin to an optical density (OD$_{600}$) of ~0.6–0.8 at 37°C and protein expression was induced with 1 mM IPTG at 17°C or room temperature. Cells were harvested after 12–18 hr by centrifugation at 5000 rpm and 4°C. FRH and CK1 were purified by standard Ni-NTA chromatography followed by ion-exchange and size-exclusion chromatography. Briefly, cells were resuspended in lysis buffer (50 mM Tris pH 8.0, 500 mM NaCl, 10% glycerol, 1 mM MgCl$_2$, 5 mM imidazole) and sonicated on ice. The lysate was clarified by centrifugation at 20,000 rpm for 1 hr. The clarified lysate was applied to a gravity Ni-NTA column and washed with five column volumes (CVs) of wash buffer (50 mM Tris pH 8.0, 500 mM NaCl, 10% glycerol, 1 mM MgCl$_2$, 20 mM imidazole). The protein was eluted using the elution buffer (50 mM Tris pH 8.0, 500 mM NaCl, 10% glycerol, 1 mM MgCl$_2$, 500 mM imidazole). The protein was then diluted into ion-exchange (IEX) buffer A (50 mM HEPES pH 7.5, 10% glycerol), applied to an S cation exchange column (GE Healthcare) and eluted over a 20 CV gradient against IEX buffer B (50 mM HEPES pH 7.5, 1 M NaCl, 10% glycerol). Fractions containing the protein were pooled, concentrated, and run over an S200 26/60 gel-filtration column pre-equilibrated in gel-filtration buffer (GFB; 50 mM HEPES, 200 mM NaCl, 10% glycerol). L-FRQ was purified under denaturing conditions. Cells were resuspended in lysis buffer (50 mM Na$_2$PO$_4$ pH 7.42, 300 mM NaCl, 5 mM imidazole, 4 M guanidinium hydrochloride [GuHCl]) and sonicated on ice. The lysate was clarified by centrifugation at 20,000 rpm for 1 hr. The clarified lysate was applied to a gravity Ni-NTA

column and washed with 3 CV of lysis buffer. The protein was eluted using the elution buffer (50 mM $Na_2PO_4$ pH 7.42, 300 mM NaCl, 500 mM imidazole, 4 M GuHCl). The protein was concentrated and applied to a S200 26/60 gel-filtration column pre-equilibrated in 50 mM $Na_2PO_4$ pH 7.42, 300 mM NaCl, 5 mM imidazole, and 4 M GuHCl. Fractions containing the protein were pooled and concentrated. The concentrated protein was flash-frozen and stored at –80°C. Prior to experiments, the purified L-FRQ was exchanged into a non-denaturing buffer (50 mM $Na_2PO_4$ pH 7.42, 300 mM NaCl, 200 mM arginine, 10% glycerol). For FFC preparations, GFB (50 mM HEPES pH 7.42, 200 mM NaCl, 10% glycerol) was used.

## SEC-MALS
Protein samples between 50 and 75 µM were injected into a Superose-6 (GE Life Sciences), equilibrated with 50 mM $Na_2PO_4$ pH 7.42, 300 mM NaCl, 200 mM arginine, and 10% glycerol. The gel filtration column was coupled to a static 18-angle light scattering detector (DAWN HELEOS-II) and a refractive index detector (Optilab T-rEX) (Wyatt Technology, Goleta, CA). Data were collected every second at a flow rate of 0.5 mL/min and analysis was carried out using ASTRA VI, yielding the molar mass and mass distribution (polydispersity) of the sample. The monomeric fraction of BSA (Sigma, St. Louis, MO) was used to standardize the light-scattering detectors and optimize data quality control.

## SEC-SAXS
X-ray scattering experiments were performed at the Cornell High Energy Synchrotron Source (CHESS) G1 station. SEC–SAXS experiments were performed using a Superdex 200 5/150 GL or Superose-6 5/150 GL (3 mL) column operated by a GE AKTA Purifier at 4°C with the elution flowing directly into an in-vacuum X-ray sample cell. Samples were centrifuged at 14,000 × $g$ for 10 min at 4°C before loading onto a column pre-equilibrated in a matched buffer (50 mM $Na_2PO_4$ pH 7.42, 300 mM NaCl, 200 mM arginine, 10% glycerol). Samples were eluted at flow rates of 0.05–0.1 mL/min. For each sample, 2 s exposures were collected throughout elution until the elution profile had returned to buffer baseline, and scattering profiles of the elution buffer were averaged to produce a background-subtracted SEC-SAXS dataset. The scattering data was processed in RAW (*Hopkins et al., 2017*) and regions with homogeneous $R_G$ values were chosen for the Guinier and Kratky analyses.

## Protein labeling for ESR spectroscopy and microscopy experiments
Sort/AEP1-tagged FRQ was cloned into a pET28a (Novagen) vector as described in *Chandrasekaran et al., 2021*. The production of the SortaseA and AEP1 enzymes and their associated peptides (GGGGC for SortaseA and CNGL for AEP1) is also described in *Chandrasekaran et al., 2021* and (*Nguyen et al., 2015*), respectively. To enzymatically label FRQ with either SortaseA or AEP1, 1:1 protein and SortaseA/AEP1 as well as 2× excess peptide were incubated overnight to allow the reaction to reach completion. The following day, the reaction was run on a Superpose 6 10/300 analytical SEC column to remove unreacted peptide and enzyme, and the protein-containing fractions were pooled and concentrated. To prepare the ADP-β-S-SL samples, ADP-β-S-SL was synthesized as previously reported in *Muok et al., 2018*. Then, 100 µL of buffer containing 100 µM of ADP-β-S-SL and 100 µM FRH or CK1 were incubated overnight and washed with GFB (50 mM HEPES pH 7.42, 200 mM NaCl, 10% glycerol) through 5× dilution followed by re-concentration to remove any unbound ADP-β-S-SL. Also, 30 µL of each sample were used for data collection. For MTSL labeling, the protein was incubated with 5× molar excess of MTSL overnight to allow the reaction to reach completion. The following day, the reaction was run on HiTrap Desalting column (GE Life Sciences) to remove any excess SL.

To prepare fluorophore-labeled FRH and CK1, the purified proteins were incubated with 2× molar excess Cy5-maleimide dye (GE Life Sciences) overnight to label any accessible cysteines on the target proteins. The excess dye was removed by running the reaction over a HiTrap desalting column equilibrated with GFB (50 mM HEPES pH 7.42, 200 mM NaCl, 10% glycerol).

## ESR spectroscopy
The cw-ESR spectra were obtained on a Bruker E500 ESR spectrometer operating at X-band (~9.4 GHz). The ESR measurements were carried out at room temperature with a modulation amplitude of 0.5–2 G and modulation frequency of 100 kHz. All spectra were digitized to 1024 points with a

sweep width of ~100 G and an average of 4–16 scans. The δ values were calculated using the display distance function on the Bruker Xepr 2.6b software.

For four-pulse DEER, the samples were exchanged into deuterated buffer containing 30% d8-glycerol, checked by cw-ESR, and plunge frozen into liquid $N_2$. The pulse ESR measurements were carried out at Q-band (~34 GHz) on a Bruker E580 spectrometer equipped with a 10 W solid-state amplifier (150 W equivalent TWTA) and an arbitrary waveform generator . DEER was carried out using four pulses (π/2- $\tau$ 1-π- $\tau$ 1-πpump- $\tau$ 2-π- $\tau$ 2-echo) with 16-step phase cycling at 60 K. The pump and probe pulses were separated by 56 MHz (~20 G). The DEER data was background subtracted with an exponential background and the distance reconstruction was carried out using denoising and the SVD method (*Srivastava et al., 2017a*; *Srivastava and Freed, 2017b*). Uncertainty in the distance distributions was calculated using the method described by *Srivastava and Freed, 2019*. Briefly, the uncertainty plotted shows the 'range' of singular values (SVs) over which the singular value decomposition (SVD) solution remains converged. For most of the data displayed, we only used the first few SVs and the solution remained converged for ±1 or 2 SVs near the optimum solution. For example, if the optimum solution was four SVs, then the range in which the solution remained converged is ~3–6 SVs. We plot three lines – lowest range of SVs, highest range of SVs, and optimum number of SVs – in the supplementary information figures; the optimum SV solution is shown in black and the region between the converged solutions with the highest and lowest number of SVs is shaded in red. Owing to the pointwise reconstruction of the distance distribution, the SVD method enables localized uncertainty at each distance value. Therefore, some points will have high uncertainty, whereas others low.

## Turbidity assays

Cysteine-less (cys-less) FRQ was created in order to minimize disulfide cross-linking at the high protein concentrations needed for in vitro phase separation experiments. For this, all four of the native cysteine residues were substituted to serine using standard molecular biology techniques. For the phase separation experiments, this cys-less FRQ variant was purified and concentrated for dialysis into phase separation buffer. Cys-less FRQ was dialyzed overnight into 25 mM Tris pH 8.0, 150 mM NaCl, or 25 mM Tris pH 8.0, 500 mM NaCl for p-FRQ and np-FRQ, respectively, at 4°C. Phase separation was induced by dialyzing in either low or high salt and increasing temperature. The protein solution was observed as a clear solution before dialysis and then turbid after dialysis, indicating that LLPS had occurred. Microscopy (see 'DIC and fluorescence microscopy') confirmed the turbid solution is due to LLPS and not aggregation. Turbidity as measured by absorbance at 300–600 nm was then recorded with an Agilent Cary 3500 UV-Vis Multicell Peltier using a temperature ramp rate of 5°C/min increasing from 10°C to 80°C. A final protein concentration of 5–15 μM was used for all turbidity assays.

## DIC and fluorescence microscopy

Phase separation was induced via overnight dialysis into 25 mM Tris pH 8.0, 150 mM NaCl, or 25 mM Tris (pH 8.0), 500 mM NaCl for p-FRQ and np-FRQ, respectively. The final protein concentration used for microscopy was 5–15 μM. Images were taken on the Zeiss Axio Imager Z1 Upright Trinocular Fluorescence Microscope at 100× magnification. To test the interaction of FRQ with FRH and CK1, the phase-separated FRQ was spiked with equimolar Cy5-labeled FRH or Cy5-labeled CK1 for final concentrations of 5 μM.

## In vitro phosphorylation assay

22 μL samples containing 10–15 μM LFRQ were incubated in the presence or absence of 2 μM CK1 and 2 μM FRH in 50 mM $Na_2HPO_4$ pH 7.5, 300 mM NaCl, 200 mM arginine, 10 mM $MgCl_2$. For the LLPS assay, FRQ was dialyzed into a buffer containing 50mM $Na_2HPO_4$ pH 7.5, 300mM NaCl and 10mM $MgCl_2$. Then, 2 μL of 2.5 mM cold ATP and 0.1 μM of [γ–32P] ATP (3000 Ci/mmol, PerkinElmer) was added to produce a total volume of ~25 μL. After ~1 hr of incubation, the reaction was quenched with 25 μL of 4× SDS with 10 mM EDTA pH 8.0 and then subjected to gel electrophoresis on a 4–20% gradient Tris-glycine gel. The gel was dried with a GelAir dryer (Bio-Rad) and placed in an imaging cassette for at least 16 hr and then imaged with a phosphorimager (GE, Typhoon FLA 7000). Band intensities were quantified using the FIJI software.

## Mass spectrometry and phospho-peptide analysis

Phosphorylated FRQ was run on a 4–20% tris-glycine SDS-PAGE gel, and the corresponding band was excised and trypsin digested. The digests were reconstituted in 0.5% formic acid (FA) for nanoLC-ESI-MS/MS analysis. The analysis was carried out using an Orbitrap Fusion Tribrid (Thermo Fisher Scientific, San Jose, CA) mass spectrometer equipped with a nanospray Flex Ion Source and coupled with a Dionex UltiMate 3000 RSLCnano system (Thermo, Sunnyvale, CA). The peptide samples (5 µL) were injected into a PepMap C-18 RP nano-trapping column (5 µm, 100 µm i.d. × 20 mm) and then separated on a PepMap C-18 RP nano column (2 µm, 75 µm × 25 cm) at 35°C. The Orbitrap Fusion was operated in positive ion mode, with spray voltage set at 1.5 kV and source temperature at 275°C. External calibration for FT, IT, and quadrupole mass analyzers was performed. In data-dependent acquisition (DDA) analysis, the instrument was operated using FT mass analyzer in MS scan to select precursor ions followed by 3 s 'Top Speed' data-dependent CID ion trap MS/MS scans at 1.6 m/z quadrupole isolation for precursor peptides with multiple charged ions above a threshold ion count of 10,000 and normalized collision energy of 30%. MS survey scans at a resolving power of 120,000 (FWHM at m/z 200) for the mass range of m/z 375–1600. Dynamic exclusion parameters were set at 35 s of exclusion duration with ±10 ppm exclusion mass width. All data were acquired using Xcalibur 4.4 operation software (Thermo Fisher Scientific). The DDA raw files with MS and MS/MS were subjected to database searches using Proteome Discoverer (PD) 2.4 software (Thermo Fisher Scientific, Bremen, Germany) with the Sequest HT algorithm. The database search was conducted against a *Neurospora crassa* database and *E. coli* database by adding LFRQ sequence into the database. The peptide precursor tolerance was set to 10 ppm and fragment ion tolerance was set to 0.6 Da with 2 miss-cleavage allowed for either trypsin or chymotrypsin. Variable modification of methionine oxidation, deamidation of asparagines/glutamine, phosphorylation of serine/threonine/tyrosine, acetylation, M-loss and M-loss + acetylation on protein N-terminus, and fixed modification of cysteine carbamidomethylation were set for the database search. Only high-confidence peptides defined by Sequest HT with a 1% FDR by Percolator were considered for confident peptide identification. All the mass spectra of phosphorylated peptides were double-checked by manual inspection.

## Integrated bioinformatic analyses and computational biology

All canonical protein and UniRef50 sequences for FRQ, FRH, and CK1, and SWISS-Prot-obtained proteome sequences of *N. crassa*, *Drosophila melanogaster,* and *Homo sapiens* were downloaded from the UniProt database (*Bateman et al., 2021*) A combination of (a) in-house-written Python scripts to allow us to analyze large datasets, (b) localCIDER (*Holehouse et al., 2017*), (c) IUPred2A (*Mészáros et al., 2018*), (d) Agadir (*Muñoz and Serrano, 1994*), (e) AlphaFold 2.0 (*Jumper et al., 2021*), (f) PScore (*Vernon et al., 2018*), and (g) catGRANULE (*Bolognesi et al., 2016*) was used to perform integrated bioinformatics analyses and computational biological studies of our proteins of interest. Unless mentioned, all programs were used by applying the default parameters. All scripts used to generate the data in this article are provided as the supplementary information datasets.

## *N. crassa* strain generation

Construction of the *frq[mNeonGreen]*, *son-1[mApple]* strain used for imaging was previously described (*Bartholomai et al., 2022b*). Briefly, a plasmid was constructed to target the endogenous *frq* locus with DNA encoding *N. crassa* codon-optimized mNeonGreen appended to the C-terminus of the *frq* ORF with a flexible linker. A gene encoding hygromycin B phosphotransferase was introduced downstream from the 3' UTR of *frq* for selection. A plasmid was constructed containing the *N. crassa son-1* ORF with mApple appended to the C-terminus with a flexible linker for insertion at the *csr-1* locus. Transformation cassettes were amplified by PCR using NEB Q5 High-Fidelity 2X Master Mix (New England Biolabs, Cat# M0492) and integrated into the genome by homologous recombination after electroporation of conidiophores. A homokaryotic strain containing the fluorescent fusion proteins was used for imaging.

## *N. crassa* growth

*N. crassa* homokaryons containing *frq[mNeonGreen]* and *csr-1::son-1[mApple]* were grown overnight on 2% agar gel pads made with Vogel's minimal medium (*Vogel, 1956*) in constant light at 25°C.

Before imaging, a section near the edge of the growing mycelium was removed and inverted on a chambered cover glass (Thermo Scientific, Cat# 155360).

### Live-cell image acquisition and processing

Images were acquired using a Zeiss AxioObserver laser scanning confocal microscope with a ×100/1.46 NA oil immersion plan-apochromat objective. 561 nm and 488 nm lasers were used at 2% power with 2× averaging and a gain of 800 for the detection of fluorescent signals by GaAsp PMTs. Images were processed for display using FIJI (*Schindelin et al., 2012*). The cropped images displaying a single nucleus were enlarged 10× using bicubic interpolation and contrasted to the minimum and maximum gray-level values. A multicolor lookup table was applied to the FRQ[mNeonGreen] channel to better observe heterogeneity of the signal, and the SON-1[mApple] channel was pseudo-colored cyan to better contrast with the lookup table applied to the FRQ[m-NeonGreen] channel. The surface plot of FRQ[mNeonGreen] was generated from the raw image. *Video 1* and *Figure 6E* were acquired using a Nikon Ti-E-based system, equipped with an Andor W1 spinning disk scan head, an Andor Zyla sCMOS camera, a 488 nm laser, and Chroma bandpass filters. *Figure 6E* is a still image from a single focal plane from a z-stack that was deconvolved using Nikon Elements and was acquired using a ×100/NA1.45 plan apo oil immersion objective. *Figure 6E* and *Video 1* were acquired using a ×60/NA1.45 plan apo oil immersion objective. The 1 min movie was acquired with a 300 ms exposure per frame. Image processing for display was carried out using FIJI.

## Acknowledgements

We thank the Cornell High Energy Synchrotron Source (CHESS) and the National Biomedical Center for Advanced ESR Technologies (ACERT) for access to data collection facilities. This work was supported by grants from the National Institutes of Health: R35GM122535 to BRC, R35GM118021 to JCD; R35GM138097 to AB, 1F31GM143890 to NM, and T32-008704 to BMB and from The Pew Charitable Trusts to AB. ACERT is supported by P41GM103521 and R24GM146107. CHESS is supported by NSF award DMR-1332208 and NIH/NIGMS award P30GM103485.

## Additional information

### Funding

| Funder | Grant reference number | Author |
| --- | --- | --- |
| National Institute of General Medical Sciences | R35GM122535 | Brian R Crane |
| National Institute of General Medical Sciences | R35GM118021 | Jay C Dunlap |
| National Institute of General Medical Sciences | R35GM138097 | Alaji Bah |
| National Institute of General Medical Sciences | 1F31GM143890 | Nicole Maurici |
| National Institute of General Medical Sciences | T32008704 | Bradley M Bartholomai |
| National Institute of General Medical Sciences | R24GM146107 | Brian R Crane |
| National Institute of General Medical Sciences | P30GM103485 | Brian R Crane |
| Pew Charitable Trusts | | Alaji Bah |

The funders had no role in study design, data collection and interpretation, or the decision to submit the work for publication.

## Author contributions
Daniyal Tariq, Conceptualization, Data curation, Formal analysis, Investigation, Methodology, Validation, Visualization, Writing – original draft; Nicole Maurici, Conceptualization, Data curation, Formal analysis, Investigation, Methodology, Validation, Visualization, Writing – original draft, Writing – review and editing; Bradley M Bartholomai, Formal analysis, Investigation, Methodology, Validation, Visualization, Writing – review and editing; Siddarth Chandrasekaran, Formal analysis, Investigation, Methodology, Visualization, Writing – review and editing; Jay C Dunlap, Conceptualization, Formal analysis, Funding acquisition, Investigation, Methodology, Project administration, Resources, Supervision, Validation, Writing – review and editing; Alaji Bah, Conceptualization, Formal analysis, Funding acquisition, Investigation, Methodology, Project administration, Resources, Software, Supervision, Writing – review and editing; Brian R Crane, Conceptualization, Formal analysis, Funding acquisition, Investigation, Methodology, Project administration, Resources, Supervision, Validation, Visualization, Writing – original draft, Writing – review and editing

## Author ORCIDs
Daniyal Tariq ⓘ http://orcid.org/0000-0002-8756-5885
Nicole Maurici ⓘ http://orcid.org/0000-0002-1463-6056
Bradley M Bartholomai ⓘ http://orcid.org/0000-0002-3082-3425
Siddarth Chandrasekaran ⓘ http://orcid.org/0000-0003-2990-7397
Jay C Dunlap ⓘ https://orcid.org/0000-0003-1577-0457
Brian R Crane ⓘ https://orcid.org/0000-0001-8234-9991

Reviewer #1 (Public Review): https://doi.org/10.7554/eLife.90259.3.sa1
Reviewer #2 (Public Review): https://doi.org/10.7554/eLife.90259.3.sa2
Reviewer #3 (Public Review): https://doi.org/10.7554/eLife.90259.3.sa3
Author Response https://doi.org/10.7554/eLife.90259.3.sa4

## Additional files

### Supplementary files
• MDAR checklist

### Data availability
The mass spectrometry FRQ phosphorylation data have been deposited to the ProteomeXchange Consortium via the PRIDE partner repository with the dataset identifier PXD037938. All other data are included in the article and/or supporting information. The raw data/scripts for all the bioinformatics analyses, biophysical experiments and imaging studies have been uploaded to Dryad and are available using the following link: https://doi.org/10.5061/dryad.pk0p2ngwh.

The following datasets were generated:

| Author(s) | Year | Dataset title | Dataset URL | Database and Identifier |
|---|---|---|---|---|
| Tariq D, Maurici N, Bartholomai BM, Chandrasekaran S, Dunlap JC, Bah A, Crane BR | 2024 | Data from: Phosphorylation, disorder, and phase separation govern the behavior of Frequency in the fungal circadian clock | https://doi.org/10.5061/dryad.pk0p2ngwh | Dryad Digital Repository, 10.5061/dryad.pk0p2ngwh |
| Tariq D, Maurici N, Bartholomai BM, Chandrasekaran S, Dunlap JC, Bah A, Crane BR | 2024 | Phosphorylation states of the Neurospora crassa circadian clock protein Frequencey | https://www.ebi.ac.uk/pride/archive/projects/PXD037938 | PRIDE, PXD037938 |

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
