## [Editor Report · eLife assessment]

This article is a **fundamental** contribution to the understanding of the role of intrinsically disordered proteins in circadian clocks and the potential involvement of phase separation mechanisms. The authors **convincingly** report on the structural and biochemical aspects and the molecular interactions of the intrinsically disordered protein FRQ. The article will be of interest to scientists focusing on circadian clock regulation, liquid–liquid phase separation, and phosphorylation.

---

## [Referee Report · Reviewer #1 (Public Review)]

Summary:

"Phosphorylation, disorder, and phase separation govern the behavior of Frequency in the fungal circadian clock" is a convincing manuscript that delves into the structural and biochemical aspects of FRQ and the FFC under both LLPS and non-LLPS conditions. Circadian clocks serve as adaptations to the daily rhythms of sunlight, providing a reliable internal representation of local time.

All circadian clocks are composed of positive and negative components. The FFC contributes negative feedback to the Neurospora circadian oscillator. It consists of FRQ, CK1, and FRH. The FFC facilitates close interaction between CK1 and the WCC, with CK1-mediated phosphorylation disrupting WCC:c-box interactions necessary for restarting the circadian cycle.

Despite the significance of FRQ and the FFC, challenges associated with purifying and stabilizing FRQ have hindered in vitro studies. Here, researchers successfully developed a protocol for purifying recombinant FRQ expressed in *E. coli*.

Armed with full-length FRQ, they utilized spin-labeled FRQ, CK1, and FRH to gain structural insights into FRQ and the FFC using ESR. These studies revealed a somewhat ordered core and a disordered periphery in FRQ, consistent with prior investigations using limited proteolysis assays. Additionally, p-FRQ exhibited greater conformational flexibility than np-FRQ, and CK1 and FRH were found in close proximity within the FFC. The study further demonstrated that under LLPS conditions in vitro, FRQ undergoes phase separation, encapsulating FRH and CK1 within LLPS droplets, ultimately diminishing CK1 activity within the FFC. Intriguingly, higher temperatures enhanced LLPS formation, suggesting a potential role of LLPS in the fungal clock's temperature compensation mechanism.

Biological significance was supported by live imaging of Neurospora, revealing FRQ foci at the periphery of nuclei consistent with LLPS. The amino acid sequence of FRQ conferred LLPS properties, and a comparison of clock repressor protein sequences in other eukaryotes indicated that LLPS formation might be a conserved process within the negative arms of these circadian clocks.

In summary, this manuscript represents a valuable advancement with solid evidence in the understanding of a circadian clock system that has proven challenging to characterize structurally due to obstacles linked to FRQ purification and stability. The implications of LLPS formation in the negative arm of other eukaryotic clocks and its role in temperature compensation are highly intriguing.

---

## [Referee Report · Reviewer #2 (Public Review)]

Summary:

This study presents data from a broad range of methods (biochemical, EPR, SAXS, microscopy, etc.) on the large disordered protein FRQ relevant to circadian clocks and its interaction partners FRH and CK1, providing novel and fundamental insight into oligomerization state, local dynamics, and overall structure as a function of phosphorylation and association. Liquid-liquid phase separation is observed. These findings have bearings on the mechanistic understanding of circadian clocks, and on functional aspects of disordered proteins in general.

Strengths:

This is a thorough work that is well presented. The data are of overall high quality given the difficulty of working with an intrinsically disordered protein, and the conclusions are sufficiently circumspect and qualitative to not overinterpret the mostly low-resolution data.

Weaknesses:

None

---

## [Referee Report · Reviewer #3 (Public Review)]

Summary:

The manuscript from Tariq and Maurici et al. presents important biochemical and biophysical data linking protein phosphorylation to phase separation behavior in the repressive arm of the Neurospora circadian clock. This is an important topic that contributes to what is likely a conceptual shift in the field.

---

## [Author Response]

The following is the authors’ response to the original reviews.

**eLife assessment**
This manuscript provides a fundamental contribution to the understanding of the role of intrinsically disordered proteins in circadian clocks and the potential involvement of phase separation mechanisms. The authors convincingly report on the structural and biochemical aspects and the molecular interactions of the intrinsically disordered protein FRQ. This paper will be of interest to scientists focusing on circadian clock regulation, liquid-liquid phase separation, and phosphorylation.
**Public Reviews:**

**Reviewer #1 (Public Review):**
Summary:"Phosphorylation, disorder, and phase separation govern the behavior of Frequency in the fungal circadian clock" is a convincing manuscript that delves into the structural and biochemical aspects of FRQ and the FFC under both LLPS and non-LLPS conditions. Circadian clocks serve as adaptations to the daily rhythms of sunlight, providing a reliable internal representation of local time.All circadian clocks are composed of positive and negative components. The FFC contributes negative feedback to the Neurospora circadian oscillator. It consists of FRQ, CK1, and FRH. The FFC facilitates close interaction between CK1 and the WCC, with CK1-mediated phosphorylation disrupting WCC:c-box interactions necessary for restarting the circadian cycle.Despite the significance of FRQ and the FFC, challenges associated with purifying and stabilizing FRQ have hindered in vitro studies. Here, researchers successfully developed a protocol for purifying recombinant FRQ expressed in *E. coli*.Armed with full-length FRQ, they utilized spin-labeled FRQ, CK1, and FRH to gain structural insights into FRQ and the FFC using ESR. These studies revealed a somewhat ordered core and a disordered periphery in FRQ, consistent with prior investigations using limited proteolysis assays. Additionally, p-FRQ exhibited greater conformational flexibility than np-FRQ, and CK1 and FRH were found in close proximity within the FFC. The study further demonstrated that under LLPS conditions in vitro, FRQ undergoes phase separation, encapsulating FRH and CK1 within LLPS droplets, ultimately diminishing CK1 activity within the FFC. Intriguingly, higher temperatures enhanced LLPS formation, suggesting a potential role of LLPS in the fungal clock's temperature compensation mechanism.Biological significance was supported by live imaging of Neurospora, revealing FRQ foci at the periphery of nuclei consistent with LLPS. The amino acid sequence of FRQ conferred LLPS properties, and a comparison of clock repressor protein sequences in other eukaryotes indicated that LLPS formation might be a conserved process within the negative arms of these circadian clocks.In summary, this manuscript represents a valuable advancement with solid evidence in the understanding of a circadian clock system that has proven challenging to characterize structurally due to obstacles linked to FRQ purification and stability. The implications of LLPS formation in the negative arm of other eukaryotic clocks and its role in temperature compensation are highly intriguing.Strengths:The strengths of the manuscript include the scientific rigor of the experiments, the importance of the topic to the field of chronobiology, and new mechanistic insights obtained.Weaknesses:This reviewer had questions regarding some of the conclusions reached.
**Recommendations For The Authors:**
The reviewer has a few questions for the authors:1. Concerning the reduced activity of sequestered CK1 within LLPS droplets with FRQ, to what extent is this decrease attributed to distinct buffer conditions for LLPS formation compared to non-LLPS conditions?

We don’t believe that these buffer conditions significantly influence the change in FRQ phosphorylation by CK1 observed at elevated temperatures. The pH and ionic strength of the buffer are in keeping with physiological conditions (300 mM NaCl, 50 mM sodium phosphate, 10 mM MgCl2, pH 7.5); CK1 autophosphorylation is robust and generally increases with temperature under these conditions (Figure 7B). However, as LLPS increases CK1 autophosphorylation remains high, whereas phosphorylation of FRQ dramatically decreases. In fact, we chose to alter temperature specifically to induce changes in phase behavior under constant buffer conditions. In this way LLPS could be increased, and FRQ phosphorylation evaluated, without altering the solution composition. Thus, we believe that the reduced CK1 kinase activity toward FRQ as a substrate is directly due to the impact of the generated LLPS milieu, i.e. the changes in structural/dynamic properties of FRQ and/or CK1 induced by the effects of being a phase separate microenvironment, which could be substantially different from non-phase separated buffer environment. For example, previous work done on the disordered region of DDX4 [Brady et al. 2017, and Nott et al. 2015] show that even the amount of water content and stability of biomolecules such as double strand nucleic acids encapsulated within the droplets differ between non- and phase separated DDX4 samples.

Nott T.J. et al. Phase transition of a disordered nuage protein generates environmentally responsive membraneless organelles. Mol. Cell. 2015 57 936-947.

Brady J.P. et al. Structural and hydrodynamic properties of an intrinsically disordered region of a germ cell-specific protein on phase separation. PNAS 2017 114 8194-8203.

In the results section we have clarified the use of temperature to control LLPS, “We compared the phosphorylation of FRQ by CK1 in a buffer that supports phase separation under different temperatures, using the latter as a means to control the degree of LLPS without altering the solution composition.”

On p.16 of the discussion we have elaborated on the above point, “We believe that the reduced CK1 kinase activity toward FRQ as a substrate is directly due to the impact of the generated LLPS milieu, i.e. the changes in structural/dynamic properties of FRQ and/or CK1 induced by the effects of being a phase separate microenvironment, which could be substantially different from non-phase separated buffer environment. For example, previous work done on the disordered region of DDX4 {Brady, 2017 #130;Nott, 2015 #131} show that even the amount of water content and stability of biomolecules such as double strand nucleic acids encapsulated within the droplets differ between non- and phase separated DDX4 samples. Indeed, the spin-labeling experiments indicate that the dynamics of FRQ have been altered by LLPS (Fig. 7D).”

1. The DEER technique demonstrated spatial proximity between FRH and CK1 when bound to FRQ in the FFC. Is there evidence suggesting their lack of proximity in the absence of FRQ? Also, how important is this spatial proximity to FFC function?

We have additional data substantiating that FRH and CK1 do not interact in the absence of FRQ. In the revised paper we have included the results of a SEC-MALS experiment showing that FRH and CK1 elute separately when mixed in equimolar amounts and applied to an analytical S200 column coupled to a MALS detector (Figure 1 below and Fig. S8). The importance of the FRH and CK1 proximity is currently unknown, but there are reasons to believe that it could have functional consequences. For example, CK1, as recruited by FRQ, phosphorylates the White-Collar Complex (WCC) in the repressive arm of the circadian oscillator [e.g. He et al. Genes Dev. 20, 2552 (2006); Wang et al, Mol. Cell 74, 771 (2019)]. Interactions between the WCC and the FFC are mediated at least in part by FRH binding to White Collar-2 [Conrad et al. EMBO J. 35, 1707 (2016)]. Thus, FRH:FRQ may effectively bridge CK1 to the WCC to facilitate the phosphorylation of the latter by the former.

He et al. CKI and CKII mediate the FREQUENCY-dependent phosphorylation of the WHITE COLLAR complex to close the Neurospora circadian negative feedback loop. Genes Dev. 2006 20, 2552-2565.

Wang B. et al. The Phospho-Code Determining Circadian Feedback Loop Closure and Output in Neurospora Mol. Cell 2019 74, 771-784.

Conrad et al. Structure of the frequency-interacting RNA helicase: a protein interaction hub for the circadian clock. EMBO J. 2016 35, 1707-1719.

**Author response image 1. sa4fig1:** Size-exclusion chromatography- multiangle light scattering (SEC-MALS) of a mixture of purified FRH and CK1. The proteins elute separately as monomers with no evidence of co-migration.

1. Is there any indication that impairing FRQ's ability to undergo LLPS disrupts clock function?

We do not currently have direct evidence that LLPS of FRQ is essential for clock function. These experiments are ongoing, but complicated by the fact that changes to FRQ predicted to alter LLPS behavior also have the potential to perturb its many other clock-related functions that include dynamic interactions with partners, dynamic post-translational modification and rates of synthesis and degradation. That said, the intrinsic disorder of FRQ is important for it to act as a protein interaction hub, and large intrinsically disordered regions (IDRs) very often mediate LLPS, as is certainly the case here. In this work, we argue that the ability of FRQ to sequester clock proteins during the TTFL may involve LLPS. Additionally, we show that the phosphorylation state of FRQ, which is a critical factor in clock period determination, depends on LLPS. Given that the conditions under which FRQ phase separates are physiological in nature and that live-cell imaging is consistent with FRQ phase separation in the nucleus, it seems likely that FRQ does phase separate in Neurospora. Furthermore, given that the sequence features of FRQ that mediate phase-separation are conserved not only across FRQ homologs but also in other functionally related clock proteins, it is probable, albeit worthy of further investigation, that LLPS has functional consequences for the clock. See the response to reviewer 3 for more discussion on this topic.

Minor Points:It might be worth noting that the cyanobacterial circadian clock also regulates LLPS, albeit involving metabolic enzymes (https://doi.org/10.1016/j.celrep.2020.108032).

Indeed, we have included a reference to this paper on p. 3: “Emerging studies in plants (Jung, et al., 2020), flies (Xiao, et al., 2021) and cyanobacteria (Cohen, et al., 2014; Pattanayak, et al., 2020) implicate LLPS in circadian clocks, and in Neurospora it has recently been shown that the Period-2 (PRD-2) RNA-binding protein influences frq mRNA localization through a mechanism potentially mediated by LLPS (Bartholomai, et al., 2022).”

On page 9, six lines from the top, please insert "of" between "distributions" and "p-FRQ".

We have corrected this typo.

**Reviewer #2 (Public Review):**
Summary:This study presents data from a broad range of methods (biochemical, EPR, SAXS, microscopy, etc.) on the large, disordered protein FRQ relevant to circadian clocks and its interaction partners FRH and CK1, providing novel and fundamental insight into oligomerization state, local dynamics, and overall structure as a function of phosphorylation and association. Liquid-liquid phase separation is observed. These findings have bearings on the mechanistic understanding of circadian clocks, and on functional aspects of disordered proteins in general.Strengths:This is a thorough work that is well presented. The data are of overall high quality given the difficulty of working with an intrinsically disordered protein, and the conclusions are sufficiently circumspect and qualitative to not overinterpret the mostly low-resolution data.Weaknesses:None
**Recommendations For The Authors:**
1)Fig.2B: Beyond the SEC part (absorbance vs elution volume), I don't understand this plot, in particular the horizontal lines. They appear to be correlating molecular weight with normalized absorption at 280 nm, but the chromatogram amplitudes are different. Clarify, or modify the plot. There are also some disconnected line segments between 10-11 mL - these seem to be spurious.

We apologize for the confusion. The horizontal lines are meant to only denote the average molecular weights of the elution peaks and not correlate with the A280 values. The disconnected lines are the light-scattering molecular weight readouts from which the horizontal lines are derived. The problematic nature of the figure is that the full elution traces and MALS traces across the peaks call for different scales to best depict the relevant features of the data. We have reworked the figure and legend to make the key points more clear.

1. It could be useful to add AF2 secondary structure predictions, pLDDT, and the helical propensity analysis to the sequence ribbon in Fig.1C.

Thank you for the suggestion, we have updated the figure to incorporate the pLDDT scores into the linear sequence map, as well as the secondary structure predictions.

1. Fig.3D: It would be better to show the raw data rather than the fits. At the same time, I appreciate the fact that the authors resisted the temptation to show distance distributions.

Yes, we agree that it is important to show the raw data; it is included in the supplementary section. Depicting the raw data here unfortunately obscures the differences in the traces and we believe that showing the data as a superposition is quite useful to convey the main differences among the sites. However, we have now explicitly stated in the figure legend that the corresponding raw data traces are given in Figures S5-6.

1. Fig.5: For all distance distributions, error intervals should be added (typically done in terms of shaded bands around the best-fit distribution). As shown, precision is visually overstated. The error analysis shown in the SI is dubious, as it shows some distances have no error whatsoever (e.g. 6nm in 370C-490C), which is not possible.

We did previously show the error intervals in the SI, but we agree that it is better to include them here as well, and have done so in the new Figure 5. With respect to the error analysis, we are following the methodology described in the following paper:

Srivastava, M. and Freed J., Singular Value Decomposition Method To Determine Distance Distributions in Pulsed Dipolar Electron Spin Resonance: II. Estimating Uncertainty. J. Phys Chem A (2019) 123:359-370. doi: 10.1021/acs.jpca.8b07673.

Briefly, the uncertainty we are plotting is showing the "range" of singular values over which the singular value decomposition (SVD) solution remains converged. For most of the data displayed in this paper we only used the first few singular values (SVs) and the solution remained converged for ± 1 or 2 SVs near the optimum solution. For example, if the optimum solution was 4 SVs then the range in which the solution remained converged is ~3-6 SVs. We plot three lines - lowest range of SVs, highest range of SVs and optimum number of SVs – in the SI figures the optimum SV solution is shown in black and the region between the converged solutions with the highest and lowest number of SVs is shaded in red. Owing to the point-wise reconstruction of the distance distribution, the SVD method enables localized uncertainty at each distance value. Therefore, some points will have high uncertainty, whereas others low. The distance that may appear to have no uncertainty has actually very low uncertainty; which can be seen at close inspection. In these cases, we observe this "isosbestic" type behavior where the P(r) appears to change little across the acceptable solutions and hence there is only a small range of P(r) values at that particular r. This behavior results from multimodal distributions wherein the change in SVs shifts neighboring peaks to lower and higher distances respectively, producing an apparent cancelation effect. What we believe is most important for the biochemical interpretation, and accurately reflected by this analysis, is the general width of the uncertainty across the distribution and how this impacts the error in both the mean and the overall skewing of the distribution at short or long distances.

Details of the error treatment as described above have been added to the supplementary methods section.

1. The Discussion (p.13) states that the SAXS and DEER data show that disorder is greater than in a molten globule and smaller than in a denatured protein. Evidence to support this statement (molten globule DEER/SAXS reference data etc.) should be made explicit.

We will make the statement more explicit by changing it to the following: “Notably, the shape of the Kratky plots generated from the SAXS data suggest a degree of disorder that is substantially greater than that expected of a molten globule (Kataoka, et al., 1997), but far from that of a completely denatured protein (Kikhney, et al., 2015; Martin, Erik W., et al., 2021). Similarly, the DEER distributions, though non-uniform across the various sites examined, indicate more disorder than that of a molten globule (Selmke et al., 2018) but more order than a completely unfolded protein (van Son et al. 2015).”

van Son, M., et al. Double Electron−Electron Spin Resonance Tracks Flavodoxin Folding, J. Phys. Chem. B 2015, 119, 13507−13514. doi: 10.1021/acs.jpcb.5b00856.

Selmke, B. et al. Open and Closed Form of Maltose Binding Protein in Its Native and Molten Globule State As Studied by Electron Paramagnetic Resonance Spectroscopy. Biochemistry 2018, 57, 5507−5512 doi: 10.1021/acs.biochem.8b00322.

1. Fig. S11B could be promoted to the main paper.

This comment makes a good point. Figure 8 is now an updated scheme, similar to the previous Fig. S11B. Thank you for the suggestion.

Minor corrections:p.1: "composed from" -> "composed of"p.2: TFFLs -> TTFLsp.2: "and CK1 via" => "and to CK1 via"p.5: "Nickel" -> "nickel"p.5: "Size Exclusion Chromatography" -> "Size exclusion chromatography"p.5: "Multi Angle Light Scattering" -> "multi-angle light scattering"Fig.2 caption: "non-phosphorylated (np-FRQ)" -> "non-phosphorylated FRQ (np-FRQ)"Fig. S3: What are the units on the horizontal axis?Fig. 5H is too smallFig. S8, S9: all distance distribution plots show a spurious "1"Fig. 6A has font sizes that are too small to readp.11: "cytoplasm facing" -> "cytoplasm-facing"p.11: "temperature dependent" -> "temperature-dependent"p.12: "substrate-sequestration and product-release" -> "substrate sequestration and product release"p.12: "depend highly buffer composition" -> "depend highly on buffer composition"

We thank the reviewer for finding these errors and their attention to detail. All of these minor points have been addressed in the revised manuscript.

**Reviewer #3 (Public Review):**
Summary:The manuscript from Tariq and Maurici et al. presents important biochemical and biophysical data linking protein phosphorylation to phase separation behavior in the repressive arm of the Neurospora circadian clock. This is an important topic that contributes to what is likely a conceptual shift in the field. While I find the connection to the in vivo physiology of the clock to be still unclear, this can be a topic handled in future studies.Strengths:The ability to prepare purified versions of unphosphorylated FRQ and P-FRQ phosphorylated by CK-1 is a major advance that allowed the authors to characterize the role of phosphorylation in structural changes in FRQ and its impact on phase separation in vitro.Weaknesses:The major question that remains unanswered from my perspective is whether phase separation plays a key role in the feedback loop that sustains oscillation (for example by creating a nonlinear dependence on overall FRQ phosphorylation) or whether it has a distinct physiological role that is not required for sustained oscillation.

The reviewer raises the key question regarding data suggesting LLPS and phase separated regions in circadian systems. To date condensates have been seen in cyanobacteria (Cohen et al, 2014, Pattanayak et al, 2020) where there are foci containing KaiA/C during the night, in *Drosophila* (Xiao et al, 2021) where PER and dCLK colocalize in nuclear foci near the periphery during the repressive phase, and in Neurospora (Bartholomai et al, 2022) where the RNA binding protein PRD-2 sequesters frq and ck1a transcripts in perinuclear phase separated regions. Because the proteins responsible for the phase separation in cyanobacteria and *Drosophila* are not known, it is not possible to seamlessly disrupt the separation to test its biological significance (Yuan et al, 2022), so only in Neurospora has it been possible to associate loss of phase separation with clock effects. There, loss of PRD-2, or mutation of its RNA-binding domains, results in a ~3 hr period lengthening as well as loss of perinuclear localization of frq transcripts. A very recent manuscript (Xie et al., 2024) calls into question both the importance and very existence of LLPS of clock proteins at least as regards to mammalian cells, noting that it may be an artefact of overexpression in some places where it is seen, and that at normal levels of expression there is no evidence for elevated levels at the nuclear periphery. Artefacts resulting from overexpression plainly cannot be a problem for our study nor for Xiao et al. 2021 as in both cases the relevant clock protein, FRQ or PER, was labeled at the endogenous locus and expressed under its native promoter. Also, it may be worth noting that although we called attention to enrichment of FRQ[NeonGreen] at the nuclear periphery, there remained abundant FRQ within the core of the nucleus in our live-cell imaging.

Cohen SE, et al.: Dynamic localization of the cyanobacterial circadian clock proteins. Curr Biol 2014, 24:1836–1844, https://doi.org/10.1016/j.cub.2014.07.036.

Pattanayak GK, et al.: Daily cycles of reversible protein condensation in cyanobacteria. Cell Rep 2020, 32:108032, https://doi.org/10.1016/j.celrep.2020.108032.

Xiao Y, Yuan Y, Jimenez M, Soni N, Yadlapalli S: Clock proteins regulate spatiotemporal organization of clock genes to control circadian rhythms. Proc Natl Acad Sci U S A 2021, 118, https://doi.org/10.1073/pnas.2019756118.

Bartholomai BM, Gladfelter AS, Loros JJ, Dunlap JC. 2022 PRD-2 mediates clock-regulated perinuclear localization of clock gene RNAs within the circadian cycle of Neurospora. Proc Natl Acad Sci U S A. 119(31):e2203078119. doi: 10.1073/pnas.2203078119.

Yuan et al., Curr Biol 78: 102129, 2022. https://doi.org/10.1016/j.ceb.2022.102129

Pancheng Xie, Xiaowen Xie, Congrong Ye, Kevin M. Dean, Isara Laothamatas , S K Tahajjul T Taufique, Joseph Takahashi, Shin Yamazaki, Ying Xu, and Yi Liu (2024). Mammalian circadian clock proteins form dynamic interacting microbodies distinct from phase separation. Proc. Nat. Acad. Sci. USA. In press.

We have updated the discussion on p. 15 accordingly:

“Live cell imaging of fluorescently-tagged FRQ proteins is consistent with FRQ phase separation in *N. crassa* nuclei. FRQ is plainly not homogenously dispersed within nuclei, and the concentrated foci observed at specific positions in the nuclei indicate condensate behavior similar to that observed for other phase separating proteins (Bartholomai, et al., 2022; Caragliano, et al., 2022; Gonzalez, A., et al., 2021; Tatavosian, et al., 2019; Xiao, et al., 2021). While ongoing experiments are exploring more deeply the spatiotemporal dynamics of FRQ condensates in nuclei, the small size of fungal nuclei as well as their rapid movement with cytoplasmic bulk flow through the hyphal syncytium makes these experiments difficult. Of particular interest is drawing comparisons between FRQ and the *Drosophila* Period protein, which has been observed in similar foci that change in size and subnuclear localization throughout the circadian cycle (Meyer, et al., 2006; Xiao, et al., 2021), although it must be noted that the foci we observed are considerably more dynamic in size and shape than those reported for PER in *Drosophila* (Xiao, et al., 2021). A very recent manuscript (Xie, et al., 2024) calls into question the importance and very existence of LLPS of clock proteins at least in regards to mammalian cells, noting that it may be an artifact of overexpression in some instances where it is seen, and that at normal levels of expression there is no evidence for elevated levels at the nuclear periphery. Artifacts resulting from overexpression are unlikely to be a problem for our study and that of Xiao et al as in both cases clock proteins were tagged at their endogenous locus and expressed from their native promoters. Although we noted enrichment of FRQmNeonGreen near the nuclear envelope in our live-cell imaging, there remained abundant FRQ within the core of the nucleus.”

**Recommendations For The Authors:**
The data in Fig 6 showing microscopy of Neurospora is suggestive but needs more information/controls. Does the strain that expresses FRQ-mNeonGreen have normal circadian rhythms? How were the cultures handled (in terms of circadian entrainment etc.) for imaging? Do samples taken at different clock times appear different in terms of punctate structures in microscopy? The authors cite the Xiao 2021 paper in *Drosophila*, but would be good to see if the in vivo picture is fundamentally similar in Neurospora.

All of the live-cell images we report were from cells grown in constant light; in the dark, strains bearing FRQ[NeonGreen] have normally robust rhythms with a slightly elongated period length as measured by a frq Cbox-luc reporter. Although we are interested, of course, in whether and if so how the punctate structures changed as function of circadian time, this is work in progress and beyond the scope of the present study. This said, it is plain to see from the movie included as a Supplemental file here that the puncta we see are moving and fusing/splitting on a scale of seconds whereas those reported in *Drosophila* by Xiao et al. (Xiao et al, 2021, above) were stable for many minutes; thus the FRQ foci seen in Neurospora are quite a bit more dynamic than those in *Drosophila*.

We have updated the results section on p. 11 to provide this information more clearly:“FRQ thus tagged and driven by its own promoter is expressed at physiologically normal levels, and strains bearing FRQmNeonGreen as the only source of FRQ are robustly rhythmic with a slightly longer than normal period length. Live-cell imaging in *Neurospora crassa* offers atypical challenges because the mycelia grow as syncytia, with continuous rapid nuclei motion during the time of imaging. This constant movement of nuclei is compounded by the very low intranuclear abundance of FRQ and the small size of fungal nuclei, making not readily feasible visualization of intranuclear droplet fission/fusion cycles or intranuclear fluorescent photobleaching recovery experiments (FRAP) that could report on liquid-like properties. Nonetheless, bright and dynamic foci-like spots were observed well inside the nucleus and near the nuclear periphery, which is delineated by the cytoplasm-facing nucleoporin Son-1 tagged with mApple at its C-terminus (Fig. 6D,E, Movie S1). Such foci are characteristic of phase separated IDPs (Bartholomai, et al., 2022; Caragliano, et al., 2022; Gonzalez, A., et al., 2021; Tatavosian, et al., 2019) and share similar patterning to that seen for clock proteins in *Drosophila* (Meyer, et al., 2006; Xiao, et al., 2021), although the foci we observed are substantially more dynamic than those reported in *Drosophila*.”

Another issue where some commentary would be helpful: Fig 7 shows that phase separation behavior is strongly temperature dependent (not biophysically surprising). Is that at odds with the known temperature compensation of the circadian rhythm if LLPS indeed plays a key role in the oscillator?

We believe that the dependence of CK1-mediated FRQ phosphorylation on temperature, as manifested by FRQ phase separation, is consistent with temperature compensation within the Neurospora circadian oscillator. The phenomenon of temperature compensation by circadian clocks involves the intransigence of the oscillator period to temperature change. Stability of period with temperature change would not necessarily be expected of a generic chemical oscillator, which would run faster (shorter period) at higher temperature owing to Arrhenius behavior of the underlying chemical reactions. Circadian phosphorylation of FRQ is one such chemical process that contributes to the oscillation of FRQ abundance on which the clock is based. Reduced CK1 phosphorylation of FRQ causes both longer periods [Mehra et al., 2009] and loss of temperature compensation (manifested as a reduction of period length at higher temperature) [Liu et al, Nat Comm, 10, 4352 (2019); Hu et al, mBio, 12, e01425 (2021)]. Thus, the ability of increased LLPS formation at elevated temperature to reduce FRQ phosphorylation by CK1 (but not intrinsic CK1 autophosphorylation) would be a means to counter a decreasing period length that would otherwise manifest in an under compensated system. As further negative feedback on the system, LLPS is also promoted by FRQ phosphorylation itself, which in turn will reduce phosphorylation by CK1. Thus, both increased FRQ phosphorylation and temperature will couple to increased LLPS and mitigate period shortening through reduction of CK1 activity.

Mehra et al., A Role for Casein Kinase 2 in the Mechanism Underlying Circadian Temperature Compensation. May 15, 2009. Cell 137, 749–760,

Liu et al. FRQ-CK1 interaction determines the period of circadian rhythms in Neurospora. Nat Comm. 2019, 10 4352.

Hu et al FRQ-CK1 Interaction Underlies Temperature Compensation of the Neurospora Circadian Clock mBio 2021 12 WOS:000693451600006.

We have added Figure 8 to clarify the interpretation of the temperature compensation implicaitons of our work, the legend of which reads:

“Figure 8: LLPS may play a role in temperature compensation of the clock through modulation of FRQ phosphorylation. Reduced CK1 phosphorylation of FRQ causes both longer periods (Mehra, et al., 2009) and loss of temperature compensation (manifested as a shortening of period at higher temperature) (Hu, et al., 2021; Liu, X., et al., 2019). Thus, the ability of increased LLPS at elevated temperature (larger grey circle) to reduce FRQ phosphorylation by CK1 will counter a shortening period that would otherwise manifest in an under compensated system. As further negative feedback, LLPS is also promoted by increased FRQ phosphorylation, which in turn will reduce phosphorylation by CK1. Thus, both increased FRQ phosphorylation and temperature favor LLPS and reduction of CK1 activity.”

one minor comment: The chemical structures in Fig 3A have some issues where the "N" and "S" are flipped. Would be good to remake these figures to fix this problem.

We apologize, the figure has been replaced with an improved version.